

# Attaining Whole-Ecosystem Warming Using Air and Deep Soil Heating Methods with an Elevated $CO_2$ Atmosphere

Paul J. Hanson[1*], Jeffery S. Riggs[2], W. Robert Nettles[1], Jana R. Phillips[1], Misha B. Krassovski[1], Leslie A. Hook[1], Lianhong Gu[1], Andrew D. Richardson[3], Donald M. Aubrecht[3], Daniel M. Ricciuto[1], Jeffrey M. Warren[1] and Charlotte Barbier[4]

[1]Climate Change Science Institute, Oak Ridge National Laboratory, Oak Ridge, Tennessee, USA.
[2]Integrated Operations Support Division, Oak Ridge National Laboratory, Oak Ridge, Tennessee, USA.
[3]Harvard University, Cambridge, Massachusetts, USA.
[4]Instrument and Source Division, Oak Ridge National Laboratory, Oak Ridge, Tennessee, USA.

*Correspondence to: P. J. Hanson, e-mail: hansonpj@ornl.gov, tel. 1-865-574-5361

Notice: This manuscript has been authored by UT-Battelle, LLC under Contract No. DE-AC05-00OR22725 with the U.S. Department of Energy. The United States Government retains and the publisher, by accepting the article for publication, acknowledges that the United States Government retains a non-exclusive, paid-up, irrevocable, world-wide license to publish or reproduce the published form of this manuscript, or allow others to do so, for United States Government purposes. The Department of Energy will provide public access to these results of federally sponsored research in accordance with the DOE Public Access Plan (http://energy.gov/downloads/doe-public-access-plan).

**Abstract.** This paper describes the operational methods to achieve and measure both deep soil heating (0-3 m) and whole-ecosystem warming (WEW) appropriate to the scale of tall-stature, high-carbon, boreal forest peatlands. The methods were developed to allow scientists to provide a plausible set of ecosystem warming scenarios within which immediate and longer term (one decade) responses of organisms (microbes to trees) and ecosystem functions (carbon, water and nutrient cycles) could be measured. Elevated $CO_2$ was also incorporated to test how temperature responses may be modified by atmospheric $CO_2$ effects on carbon cycle processes. The WEW approach was successful in sustaining a wide range of above and belowground temperature treatments (+0, +2.25, +4.5, +6.75 and +9 °C) in large 115 $m^2$ open-topped chambers with elevated $CO_2$ treatments (+0 to +500 ppm). Air warming across the entire 10 enclosure study required ~90% of the total energy for WEW ranging from 64283 MJ $d^{-1}$ during the warm season to 80102 MJ $d^{-1}$ during cold months. Soil warming across the study required only 1.3 to 1.9 % of the energy used ranging from 954 to 1782 MJ $d^{-1}$ of energy in the warm and cold seasons, respectively. The residual energy was consumed by measurement and communications systems. Sustained temperature and elevated $CO_2$ treatments were only constrained by occasional high external winds. This paper contrasts the in situ WEW method with closely related field warming approaches using both above (air or infrared heating) and belowground warming methods. It also includes a full discussion of confounding factors that need to be considered carefully in the



interpretation of experimental results. The WEW method combining aboveground and deep soil
heating approaches enables observations of future temperature conditions not available in the
current observational record, and therefore provides a plausible glimpse of future environmental
conditions.



## 1. Introduction

Measurements through time and across space have shown that the responses of terrestrial
ecosystems to both chronic and acute perturbations of climatic and atmospheric drivers can lead
to changes in ecosystem structure (e.g., species composition, leaf area, root distribution; IPCC
2014, Walther et al. 2002, Cramer et al. 2001) and ecosystem function (e.g., plant physiology,
soil microbial activity, and biogeochemical cycling; Bronson 2008, 2009). The projected
magnitudes and rates of future climatic and atmospheric changes, however, exceed conditions
exhibited during past and current inter-annual variations or extreme events (Collins et al. 2013),
and thus represent conditions whose ecosystem-scale responses may only be studied through
manipulations at the field scale. Science working groups have focused on next generation
ecosystem experiments (Hanson et al. 2008) and concluded that there is "a clear need to resolve
uncertainties in the quantitative understanding of climate change impacts" and that "a
mechanistic understanding of physical, biogeochemical, and community mechanisms is critical
for improving model projections of ecological and hydrological impacts of climate change."
Furthermore, a number of reviews have recently called for new studies of climate extremes
including experimental warming to obtain measurements for warming scenarios that go beyond
the observable records (Cavaleri et al. 2015; Kayler et al. 2015; Torn et al. 2015).

Consensus projections of the climatic and atmospheric changes from the Fifth Assessment
Report of the Intergovernmental Panel on Climate Change (IPCC) vary spatially across the
globe. Warming is, however, projected to be greatest at high latitudes with temperature increases
larger in winter than summer (Collins et al. 2013). A mean warming of as much as 2.6 to 4.8°C
during the summer and winter respectively is expected by the end of this century, based on GCM
calculations for the IPCC RCP8.5 scenario. That level of warming exceeds the typically
observed variation in mean annual temperatures (±2°C) and therefore represents a range of
conditions that necessitates experimental manipulation. In addition, future extreme summer heat
events may expose ecosystems to acute heat stress that exceed historical and contemporary long-
term conditions for which extant vegetation is adapted.

Warming has been studied using many methods in field settings with the most common methods
focused on warming low stature or juvenile vegetation and surface soils using infrared heaters,



small open top chambers or near-surface heating cables - all of which have restricted warming
capacities (Aronson and McNulty 2009). This paper describes warming methodologies that take
us to the other extreme: systems capable of producing warming at multiple temperature levels in
larger plots (>100 m$^2$) and throughout the soil profile (depths well below 1 m) and above tall
vegetation. The methodology was initially demonstrated in a small 12 m$^2$ chamber (Hanson et al
2011), scaled up to a full-sized prototype >100 m$^2$ (Barbier et al. 2012), then deployed into a
black spruce – sphagnum peat bog in northern Minnesota as a platform for the Spruce and
Peatland Response Under Climatic and Environmental Change (SPRUCE) experiment
(http://mnspruce.ornl.gov; Krassovski et al. 2015)

SPRUCE was conceived to provide whole-ecosystem experimental treatments that span a wide
range of warming scenarios to improve understanding of mechanistic processes and
consequential ecosystem-level impacts of warming on peatlands. SPRUCE is evaluating the
response of existing *in situ* and tall-stature (>4 m) biological communities to a range of
temperatures from ambient conditions to +9°C for a *Picea mariana* (Mill.) B.S.P. [black spruce]
– *Sphagnum* spp. peatland forest in northern Minnesota. Because this ecosystem is located at the
southern extent of the spatially expansive boreal peatland forests it is considered to be especially
vulnerable to climate change, and warming is expected to have important feedbacks on the
atmosphere and climate through enhanced greenhouse gas emissions (Bridgham 2006; Davidson
and Janssens 2008; Strack et al. 2008). The primary goals of the research were 1) to test how
vulnerable an important C-rich terrestrial ecosystem is to atmospheric and climatic change, 2) to
test if warming of the entire soil profile would release large amounts of $CO_2$ and $CH_4$ from a
deep C-rich soil, and 3) to derive key temperature response functions for mechanistic ecosystem
processes that can be used for model validation and improvement. SPRUCE provides an
excellent opportunity to investigate how atmospheric and climatic change alter the interplay
between vegetation dynamics and ecosystem vulnerability, while addressing critical uncertainties
about feedbacks through the global C and hydrologic cycles.

This paper describes the operational methods applied to achieve both deep soil heating, or in this
case, deep *peat* heating (DPH), and whole-ecosystem warming (WEW) appropriate to the scale
of the 6-m tall boreal forest and underlying peat. While the primary goal for SPRUCE was to



focus on the response of a high-C peatland ecosystem to rising temperatures, elevated $CO_2$
($eCO_2$) was also incorporated into the experimental design to test how the temperature response
surfaces may be modified by expected changes in atmospheric $[CO_2]$. The paper further
describes confounding factors that need to be considered carefully in the interpretation and
analysis of the experimental results (Leuzinger et al. 2015). While a comprehensive literature
comparison to other warming methods (Rustad et al. 2001; Shaver et al. 2000; Aronson and
McNulty 2009) was not an objective of this paper, the nature of the *in situ* WEW method is
discussed in the context of closely related field warming approaches deployed with both above
(air or infrared heating) and belowground warming methods.

**2.  Methods**
**2.1 A brief discussion of the SPRUCE Experimental Infrastructure**
Experimental plots and infrastructure in support of the SPRUCE WEW study were established
on the S1-Bog of the Marcell Experimental Forest (MEF; Kolka et al. 2011). Raised boardwalks
were added in 2012, electrical and communication system were added in 2013, provisions for
belowground heating were added in 2014, and the aboveground enclosures and air warming
systems were added between January and June of 2015. Infrastructure for the addition of $eCO_2$
was added in 2016. Pretreatment data were collected throughout the 2012 to 2015 period.

An original plan for the SPRUCE experimental temperature and $CO_2$ treatments included a
traditional replicated ANOVA design, but a quantitative analysis of various experimental designs
and discussions among experimentalists and modelers led to the conclusion that a regression-
based experimental design (Cottingham et al. 2005) including a broad range of temperature
levels would yield long-term data more suited for the characterization of response curves for
application within ecosystem and earth system models (see also Kardol et al. 2012). If necessary
for some assessments of significance warming effects (e.g., individual tree growth), the
regression combination of treatment plots might be justifiably binned into low, medium and high
temperature treatments for ANOVA-based analyses. An important assumption underlying this
choice was that there were no strong gradients across the experimental area that would mandate a
block design. Preliminary survey data from the chosen site justify making this assumption (e.g.,
Parsekian et al. 2012; Tfaily et al. 2014).





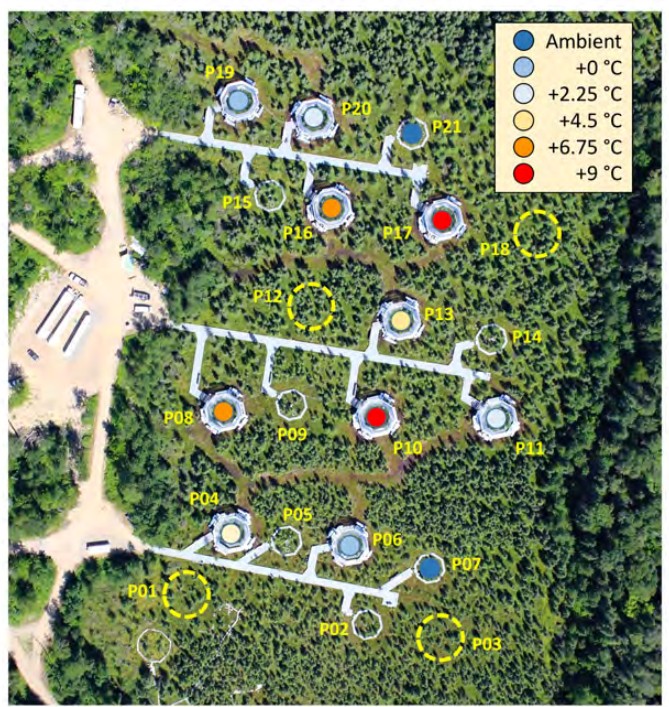

**Figure 1**: Aerial photograph of the SPRUCE experimental site on August 5, 2015. Plot numbers
(1 to 21) and assigned temperature treatments are superimposed on the image. Dashed circles
indicated established plot centers for plots that are monitored annually for tree growth. Plots 4,
10, 11, 16 and 19 receive elevated $CO_2$. The middle boardwalk is 112 m long.
An aerial photograph of the SPRUCE site shows the random assignment of treatments to plots
(Fig. 1). Tfaily et al. (2014) and Krassovski et al. (2015) provide details for the experimental site,
which include three ~100 m transect boardwalks for accessing 17 octagonal permanent plots
over the southern half of the 8.1 ha bog. Electrical supply systems (for belowground heating and
instrumentation), propane vaporizers and delivery pipelines (for forced-air heating), pure $CO_2$
delivery pipelines (for e$CO_2$ additions), and a data communication network (Krassovski et al.
2015) were initially installed along each transect to serve the individual permanent plots. Ten of
the permanent plots were randomly assigned to the following warming treatments: 2 fully-
constructed control plots with no energy added (henceforth simply control plots), and 2 plots
each to be managed as +2.25, +4.5, +6.75 and +9 °C warming plots. Two unchambered ambient





plots are also part of the experimental design. Enclosure methods for warming of the air and
belowground peat are described further in the following sections.

Each of the ten plots is surrounded beneath the surface by a corral made of interlocking vinyl
sheet pile walls (Model ESP 3.1, EverLast Synthetic Products, LLC) for the hydrologic isolation
of each plot as an independent ombrotrophic system (Sebestyen and Griffiths 2016). Following
installation, each sheet piles extends above the bog surface approximately 0.3 m having been
driven vertically through the peat profile (3 to 4 m) into the underlying ancient lake sediment.
Slotted outflow pipes allow for lateral drainage and hydrologic measurements and sampling from
each plot. The operation and performance of the corral system will be described in a future
paper. During the period of performance covered in this manuscript, the bog remained very wet
with a water table near the surface.

**2.2 Site Description**
The climate of the MEF is subhumid continental, with large and rapid diurnal and seasonal
temperature fluctuations (Verry et al., 1988). Over the period from 1961 through 2005 the
average annual air temperature was 3.3 °C, with daily mean extremes of -38 °C and 30 °C, and
the average annual precipitation was 768 mm. Mean annual air temperatures have increased
about 0.4 °C per decade over the last 40 years (Sebestyen et al., 2011).

The investigated peatland is the S1-Bog of the MEF (N 47° 30.476'; W 93° 27.162' and 418 m
above mean sea level). The S1-Bog is an ombrotrophic peatland with a perched water table that
has little regional groundwater influence. The S1-Bog is dominated by *Picea mariana* (Mill.)
B.S.P. (black spruce) with contributions to the forest canopy from *Larix laricina* (Du Roi) K.
Koch (larch). The S1-Bog was harvested in strip cuts in 1969 and 1974 to test the effects of
seeding on the natural regeneration of *P. mariana*. In its current state of regeneration, the canopy
is 5-8 m tall. The peatland soil is the Greenwood series, a Typic Haplohemist
(http://websoilsurvey.nrcs.usda.gov) with average peat depths to the Wisconsin glacial-age lake
bed of 2 to 3 m (Parsekian et al., 2012). Recent surveys of the peat depth, bulk density, and C
concentrations for the S1-Bog suggest a total C storage pool of greater than 240 kgC m$^{-2}$





(calculated to a 3 m average depth), with greater than 90% over 3000 years old (Karis
McFarlane, personal communication).

Vegetation within the S1-Bog is dominated by two tree species (see above), and is supported by
a bryophyte layer dominated by *Sphagnum* spp mosses, especially *S. angustifolium* and *S. fallax*
in hollows and *S. magellanicum* on drier hummocks. Other mosses including *Pleurozium* spp
(feather mosses) and *Polytrichum* spp (haircap mosses) are also present. The understory includes
a layer of ericaceous shrubs including *Rhododendron groenlandicum* (Oeder) Kron & Judd
(Labrador tea), *Chamaedaphne calyculata* (L.) Moench. (leatherleaf) with a minor component of
other woody shrubs. The bog also supports a limited number of herbaceous species including:
the summer-prevalent *Maianthemum trifolium* (L.) Sloboda (Three-leaf false Solomon's seal), a
variety of sedges (*Rhynchospora alba* (L.) Vahl, *Carex* spp.) and *Eriophorum vaginatum* (cotton
grass). The belowground peat profile and geochemistry are described in Tfaily et al. (2014).

**2.3 Air warming protocols**
Air warming was achieved by heating the air above the surface of the peatland to a height of
nearly 6 m within open top octagonal enclosures (7 m tall by 12.8 m in diameter with an area of
114.8 $m^2$; Fig. 2A). The enclosures include an octagonal open top (8.8 m diameter with an area
of 66.4 $m^2$) bounded by a 35° frustum. The frustum was added to enhance the efficiency of the
warming enclosure (Barbier et al. 2012). Wall and frustum structural members were made of
structural aluminum (6061-T6 Alloy), and the walls are sheathed with double walled transparent
greenhouse panels (16 mm acrylic glazing). The vertical walls of the enclosure sit approximately
0.46 m above the bog hollow surface. The gap from the bottom of the enclosure was sealed into
the bog surface (~10 cm) with flexible acrylic panels. All structures are supported above the bog
on helical piles installed to a typical depth of 12 to 18 m below the peat surface within stable
ancient lake sediments and glacial till.




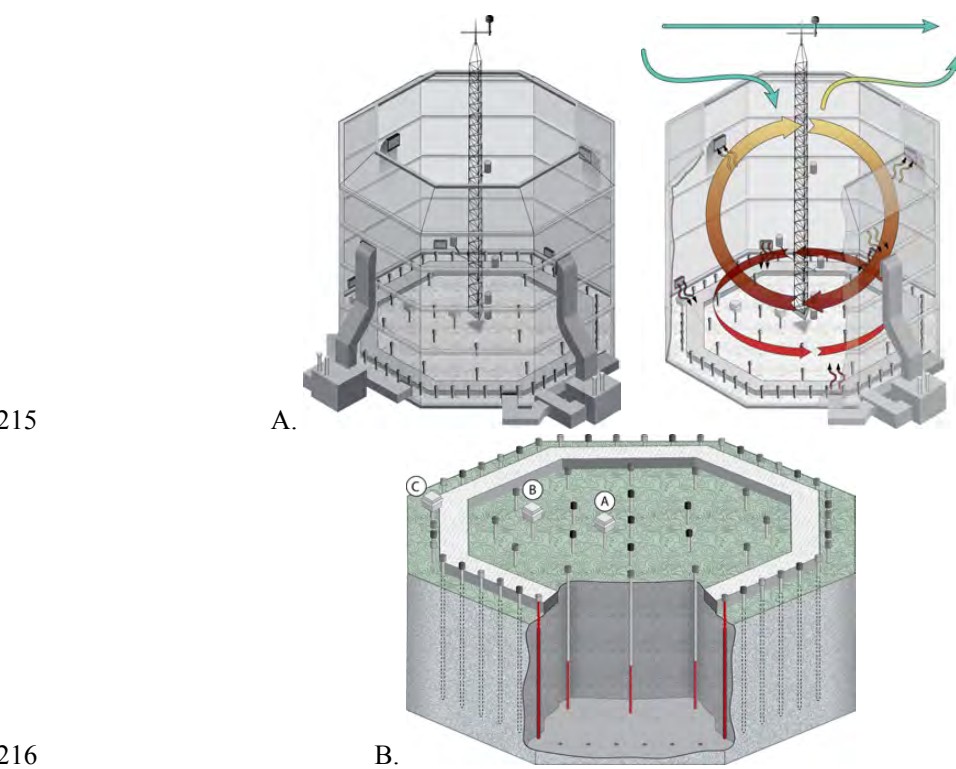

215       A.

216       B.
**Figure 2:** Panel A: Diagram of the air warming enclosure, warm air flow pattern, and external wind inputs leading to a homogenized air envelope that surrounds the aboveground vegetation. Panel B: Diagram of the belowground heater distribution pattern and the functional heating surfaces. The 100 W heaters are deployed in an inner section A (7 deep only heaters), B (12 deep only heaters), and C (three alternating circuits of 48 full length heaters).

Air warming method theory, protocols and optimization of an earlier prototype were fully
described by Barbier et al. (2012). Briefly, air at four mid-enclosure heights was drawn from
within the enclosure down to four ground level propane indirect fired bent tube heaters (Model
A2-IBT-600-300-300-G15; CaptiveAire, Youngsville, NC)) for variable heating of the air to
achieve five temperature targets (+0, +2.25, +4.5, +6.75 and +9 °C). The pattern of air flow and
air warming within a typical enclosure is depicted in Fig. 2A. Warmed air from the 4 heat
exchangers is split into eight equal distribution conduits for distribution into the enclosure 1
meter above the peat hollow surface through diffusers located on each wall. The control or warm
air delivered into each enclosure is provided at a continuous mean velocity of 7.5 m s$^{-1}$ (blower
operation was initiated in 2015 as soon as each enclosure was fully glazed with greenhouse





panels). These warm air streams are directed away from adjacent vegetation surfaces as much as
possible and diffuse rapidly into the background mixed air of the enclosure.

The air warming described above was achieved using propane fired heat exchangers. Propane
was delivered to a large (10000 gallon) liquid propane storage tank located at the site. Liquid
propane was pulled from the bottom of this tank and pumped to vaporizers located at the head of
each boardwalk. Vaporized propane was then piped to the furnaces. This system allowed us to
operate throughout the year including periods of ambient winter temperatures as low as -35 °C
on January 17, 2016.

**2.4 Peat warming protocols**
In June of 2014 when the capabilities for deep belowground warming were operational, we
initiated a 13-month period of DPH treatments for the 10 constructed SPRUCE plots. The DPH
method is an expanded form of the deep belowground heating approach of Hanson et al. (2011)
that was rationalized as being an appropriate surrogate for deep soil heating expected under
future climate conditions (Huang 2006; Baker et al. 1993). DPH was accomplished by an array
of 3-m vertical low wattage (100 W) heating elements installed throughout the plots within a
plastic-coated iron pipe. The belowground heating array, which was contained within the
encircling subsurface corral, included circles of 48, 12, and 6 heaters at 5.42, 4 and 2 m radii,
respectively (Fig. 2B). A single heater was also installed at the plot center. Exterior heaters in the
circle of 48 applied the 100 W across the full linear length of the heater, and all interior heaters
applied their 100 W heating capacity to the bottom one third of each resistance heater (pipe
thread core heaters, Indeeco, St. Louis, MO). Interior heaters were different to avoid directly
heating the peat volumes targeted for the measurement of response variables.

**2.5 Temperature Control**
Simple proportional-integral-derivative (PID) control was used for aboveground heating based
on differentials measured by duplicate sensors in the center of the plot at +2 m. In the each above
ground heating system, the position of a liquid petroleum gas (LPG) valve in each of the four
heating units was simultaneously controlled. The belowground heating system controlled
individual heating circuits with silicon controlled rectifiers (SCR Controller: 1 Phase, 1 Leg.



240V, 20 Amb @42.5 °C; 4-20 mA control, Watlow Model DA10-24-F0-0-00) in each of 5
circuits. DPH within the experimental plots was achieved through PID control of three exterior
(the circle of 48 split into alternating thirds) and two interior circuits of the resistance heaters
shown in Fig. 2B. The control depth was -2 m. The reference for air and belowground heating
was the Plot 06 control plot. Details for above and belowground PID control are provided in the
supplemental materials to this paper along with PID coefficients for each warming treatment.

**2.6 Elevated CO$_2$ Additions**
Logical projections from IPCC analyses and the recent evaluation of current emissions (Raupach
et al. 2007; Collins et al. 2013) suggests that experimental methods might consider atmospheric
CO$_2$ concentrations at or above 800 ppm based on current fossil fuel use. As with the warming
targets, the goal of the SPRUCE infrastructure was not to simulate a specific future climate or
atmospheric condition, but to include a [CO$_2$] representative of the high end of predicted values
for the end of the century (Collins et al. 2013). The eCO$_2$ additions were included to better
understand the potential mechanism that CO$_2$-induced enhancements of gross primary production
might have on warming responses.

Pure CO$_2$ additions were initiated in half of the treatment plots (one for each temperature
manipulation) on 15 June 2016 to provide an eCO$_2$ atmosphere approaching 900 ppm (nominally
+500 ppm over current conditions in 2016) during daytime hours. The selected value is
purposefully higher than concentrations used in previous large eCO$_2$ experiments (Medlyn et al.
2015), and might be expected to yield a greater response by the trees and shrubs of the S1-Bog.
The following text briefly describes the mechanism for elevating CO$_2$ within the WEW
enclosures. Half-hourly assessments of [CO$_2$] in air were obtained at 0.5, 1, 2 and 4 m by
continuously sampling of air from plot-center tower locations via a sampling manifold.
Individual elevations were sampled in series for 90 seconds over a 6 minute cycle. The sampled
gas stream was analyzed using an in line LiCor LI-840 CO$_2$/H$_2$O gas analyzer at a flow rate of 1
L min$^{-1}$.

The presence of the enclosure walls reduces air turnover within the experimental space and limits
the amount of CO$_2$ needed as compared to Free-Air CO$_2$ Enrichment (FACE) studies (e.g.,



Dickson et al. 2000). Source $CO_2$ for the SPRUCE experiment was obtained from a fossil-fuel
based fertilizer plant by the contracted $CO_2$ supplier (Praxair, Inc.) and has $\partial^{13}C$- and $\Delta^{14}C$-$CO_2$
signatures of ~54 ‰ and -1000 ‰, respectively. Pure $CO_2$ from a central storage area (two 60-
ton refrigerated tanks) is vaporized and transferred by pipeline to each enclosure where it is
warmed and regulated before entering a mass flow control valve (model GFC77, 0-500 LPM
$CO_2$, 4-20 mA control; Aalborg Instruments and Controls, Inc.). The mass flow control valve
allows for variable additions of the pure $CO_2$ to the enclosure. A typical delivery velocity for
pure $CO_2$ equals 250 L min$^{-1}$, but ranges from 100 to 500 L min$^{-1}$ with external wind velocities
between 0.2 and 5 m s$^{-1}$ to account for increasing air volume turnover. Warm air buoyancy
increases with larger temperature differentials (Barbier et al. 2012) and increases air turnover
rates and demands for $CO_2$ additions.

The enclosure's regulated additions of pure $CO_2$ are distributed to a manifold that splits the gas
into four equal streams feeding each of the 4 air handling units (Fig. 2A), and is injected into the
ductwork of each furnace just ahead of each blower and heat exchanger. Horizontal and vertical
mixing within each enclosure homogenizes the air volume distributing the $CO_2$ along with the
heated air. Details of the $CO_2$ addition algorithms as they are impacted by external winds are
provided in the supplemental materials.

**2.7 Bog and Enclosure Environmental Measurements**
Half-hourly mean air temperature measurements were made with thermistors (Model HMP-155;
Vaisala, Inc.) installed at the center of each plot at 0.5, 1, 2 and 4 meters above the surface of the
peat. These same sensors included a capacitance sensor for the measurement of relative
humidity. New or recalibrated sensors are deployed annually or as comparisons to other sensors
suggest failure. Multipoint thermistor probes for automated mean half-hour peat temperature
measurements (W.H. Cooke & Co. Inc, Hanover, PA) were custom designed from a 1.3 cm
diameter x 0.9 mm wall stainless steel tube with a 7.62 cm stainless steel disk welded at the zero
height position along the tube. All elevations within the bog are referenced to the peat surface
hollows, which are defined to be an elevation of 0 cm. An electrical termination enclosure was
supported above the bog surface by a 46 cm extension of the measurement tube to avoid shading
the bog surface at the point of measurements and to keep it above any standing water. Peat





temperatures were recorded at 9 depths for the designated experimental plots (0, -5, -10, -20, -30,
-40, -50, -100 and -200 cm) at three concentric zones (one at 5.42-m radius; one at 3-m radius;
one at 1-m radius; Fig. 2B). All integrated temperature probes were located at a midpoint
between heaters in a given concentric ring of the plot. Hummock temperature measurements
were also obtained in the hummocks at various elevations above the hollow surface
(approximately 0, +10, and +20 cm).

Photosynthetically active radiation (PAR) was measured with quantum sensors (LiCor Inc., LI-
190R) at 2.5 m above the surface at a middle plot location. Supplemental 1-min short wave
(pyranometer, 300 to 2800 nm) and long wave (4.5 to 42 µm) radiation observations were also
measured using matched net radiometers (Model CNR4, Kipp and Zonen) for unchambered
ambient and within-enclosure locations for selected mid-summer days to further characterize the
enclosure environment.

Soil water content is difficult to measure in heterogeneous, low density organic soils.
Nevertheless, volumetric water content was measured within hummocks at two depths (0 cm at
the hollow surface, and 20 cm below hummock surface) at three locations within each plot using
a capacitance/frequency domain sensor (10HS, Decagon Devices Inc.). These sensors required
site-specific calibration (Supplemental Fig. S1).

External wind sensors at + 10 m above the center of each enclosure (Windsonic 4; Gill
Instruments) provided important information necessary to estimate the mixing of ambient air into
the enclosure space. A mobile 3-D sonic anemometer (Campbell Scientific Inc., Logan, Utah;
Model CSAT3B) was also temporarily deployed inside and outside of Plot 6 to characterize the
nature of turbulence changes inside and outside of the enclosures.

**2.8 Image collections**
Infrared imaging of the internal air space was done periodically to evaluate the spatial pattern of
heating of biological surfaces within the warming enclosures. Images were collected with a
thermal imaging camera (TiR4 #2816061, Fluke Corporation, Everett, WA) with a 20mm F/0.8
8-14 µm lens. Images were taken at the entrance of each enclosure (or unchambered ambient



space) immediately after the door was opened. All images in a comparative series were collected
before or after sunset within 20 minutes of one another (the time it takes to move about the
SPRUCE site).

Whole-plot visible wavelength image cameras (StarDot NetCam SC Series SD130BN 1.3MP
MJPEG Hybrid Color Day/Night IP Box Camera with 4mm Lens) were installed as a part of the
PHENOCAM network (Keenan et al. 2014; Toomey et al. 2015). These cameras provide a view
of the entire enclosure area. The whole plot imaging cameras record visible (400-700 nm) and
visible plus infrared (400-1000 nm) images sequentially, allowing calculation of NDVI-type
indices (Petach et al. 2014). They are installed on the southern wall of each enclosure at a height
of 6 m. Current and archived PHENOCAM images for the SPRUCE plots can be found at
https://phenocam.sr.unh.edu/webcam/gallery/.

**2.9 Energy Balance modeling**
The energy balance in the S1 bog both inside and outside the enclosures was simulated using the
Community Land Model (CLM) version 4.5 (Oleson et al., 2013), which was modified to
represent the specific hummock-hollow microtopography, runoff and subsurface drainage at the
S1-Bog (Shi et al., 2015). This CLM-SPRUCE model was driven by meteorological data
collected by the environmental monitoring stations in the S1-Bog between 2011 and 2015.
Enclosure impacts on both incoming longwave and shortwave radiation were also considered in
the simulations. The incoming longwave radiation at the surface within an enclosure is estimated
by assuming the enclosure walls emit blackbody radiation at a temperature equal to the simulated
2-meter air temperature, and by using a sky view factor (defined as the proportion of the
longwave radiation received by the surface within the enclosure that comes from the clear sky)
of 0.3 to 0.35. The sky view factor is assumed to be 1 outside the enclosure (neglecting the
effects of the vegetation itself), while the inside values are calculated using the enclosure
geometry. The enclosure walls are also assumed to cause a 20% reduction in incoming
shortwave radiation. For these simulations, we do not consider the impacts of the enclosures on
wind speed, precipitation, or pressure. The effects of the enclosures on air and vegetation
temperature, snow cover, dew formation and energy fluxes are simulated by the model and
reported in the Discussion (Section 4).





# 3. Results

## 3.1 Warming Differentials

WEW in the S1-Bog was achieved by warming air throughout the vertical profile of tall vegetation within an open topped enclosure combined with belowground warming using low-wattage electrical resistance heaters optimized to the 12-m diameter space. Figure 3 demonstrates the effectiveness of the belowground heating method to produce a consistent deep soil (peat) warming at –2 m beginning in the summer of 2014. Peat is also warmed below -2 m,

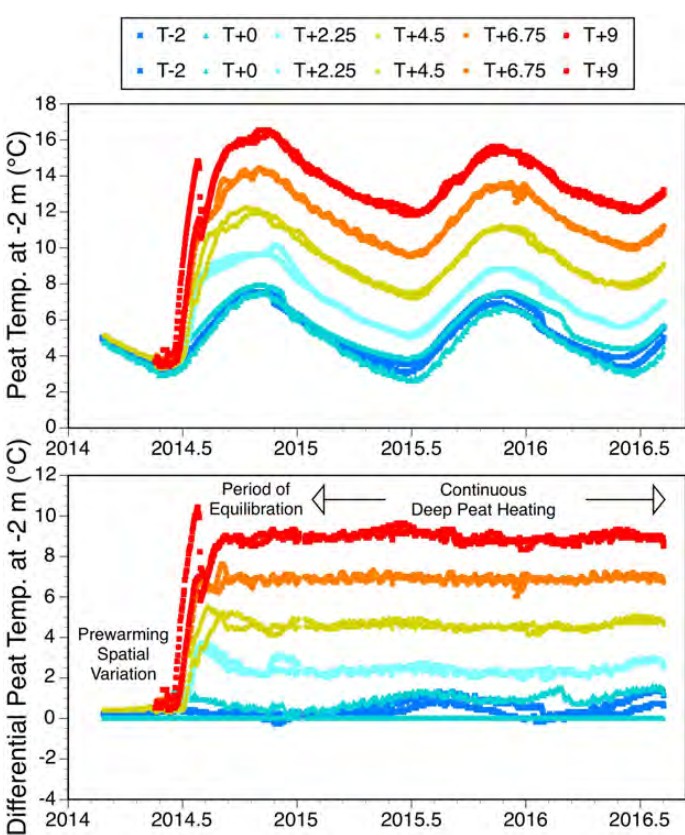

**Figure 3:** Daily mean deep peat temperatures (A) and the associated temperature differentials (B) at -2 m by treatment plots since 2014 including the initial warm up periods (June through early September 2014), and the sustained application of deep peat heating with air warming (beginning September 2014). Differential temperatures are referenced to sensors within the fully constructed but no-energy-added control Plot #6. Unchambered ambient plot data are also shown as T-2 plots.



but continuous temperature monitoring below the -2 m zone was not done. Differential deep soil
temperature targets were sustained following periods of gradual heat accumulation from 22 to 94
days for the cooler and warmest treatments respectively (see Supplemental Table S3). Once deep
soil temperatures were achieved they were maintained consistently through time with the
exception of a few minor power interruptions or during instrument maintenance periods. Deep
soil temperatures in unchambered ambient plots (T-2 lines in Fig. 3) were warmer than the
designated reference control plot (Plot 6). Variation in the no-energy-added controls (Plot 6
versus Plot 19) represented spatial differences that were likely driven by variation in tree canopy
cover.  Greater canopy cover (Plot 19) leading to warmer peat temperatures due to less heat loss
to the sky.

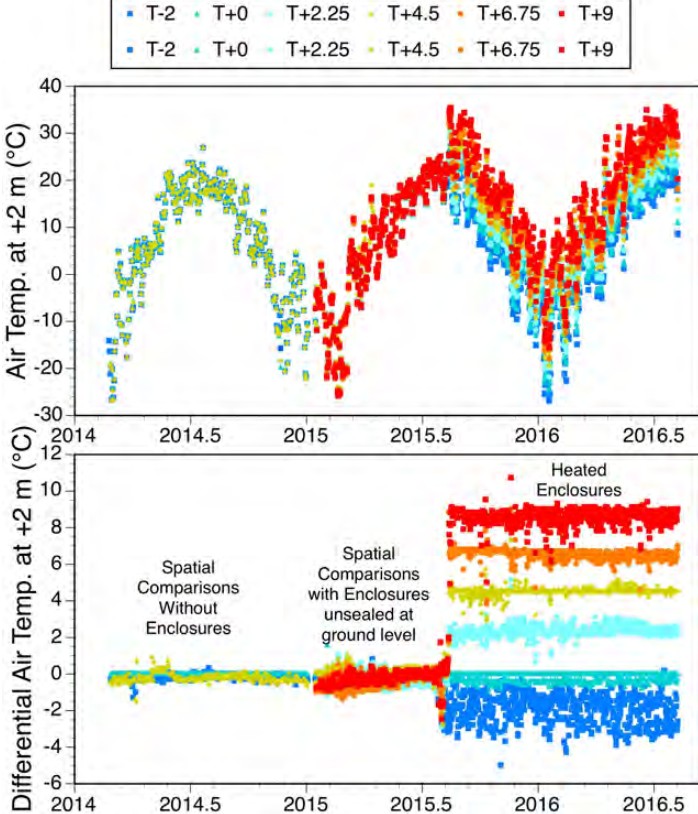

**Figure 4:** Daily mean air temperatures (A) and the associated air temperature differentials at +2
m above the bog surface by treatment plots since 2014 including periods prior to enclosure
construction (through January 2015), a period when upper enclosures were in place (January to
July 2015), and observations since full enclosure of each plot was achieved (27 July through 5
August 2015). Interior blower function was initiated at the time of full plot enclosure. The



sustained period of warming began at 14:00 on 12 August 2015. Differential temperatures are
referenced to sensors within the fully constructed but no-energy-added control Plot #6.
Unchambered ambient plot data are also shown as T-2 plots.

Figure 4 shows consistent pretreatment seasonal air temperature patterns across plots prior to the
full enclosure of the warming plots. Enclosure installations minus the bottom row of glazing was
completed between mid-January and early April of 2015. During the period from April through
July 2015 air handling units and duct work were installed. The bottom row of glazing was added
in mid-August 2015 followed immediately by the initiation of constant stirring of the internal air
space by the recirculating air handling furnaces. Air warming was initiated in all plots on August
12, 2015, and has been maintained near target levels since that time unless power outages or
system maintenance needs interrupted operation (Fig. 4).
Unchambered ambient plots are commonly from 1 to 3 °C cooler than the fully constructed
controls (Fig. 4), and plot to plot variation is responsible for the difference between our Plot 6
reference control and Plot 19 (the other no-energy-added control plot). The system based on PID
control of 2 m air temperatures at the center of each enclosure is routinely capable of maintaining
the differential temperatures for the +2.25 and +4.5 plots under virtually all environmental
conditions. Currently, at higher winds (> 3 m s$^{-1}$) and for short periods of time the system
occasionally falls below the +6.75 and +9 °C target temperatures (especially in the +9 °C Plots
#10 and 17). We continue to work on adjustments to the PID settings to minimize such issues,
which are driven by the dilution of internal warm air by atypical cold air intrusions through the
enclosures open top.

Since the initiation of DPH on July 2, 2014 belowground warming has been actively engaged
greater than 98 % of the time for all plots except Plot 11 which was operated 93% of the time
(Table 1). Because the deep soils are largely self-insulated, downtime for active DPH
management resulted in only minor deviations from the target temperatures (Fig. 3). Active
aboveground warming, initiated on August 13, 2015, has been maintained greater than 99 % of
the time in 7 of 8 plots and more than 96.5 % of the time in Plot 11. When aboveground heating
fails for any reason, differential heating is lost almost immediately adding air temperature
variations greater than present in other plots that have not failed. Plot 11 downtime was the result



of Transect 2 power outages and winter issues with the air warming heat exchangers (i.e.,
furnaces). Table 1 provides further details on the percent of days in which the mean temperature
was within 0.2, 0.5, 1 or 1.5 °C of the established targets for a given treatment plot.

**Table 1.** Statistics for time of operation and time within operational target temperature ranges for
each treatment enclosure or plot. (**A**) Percent of time for active deep peat heating (DPH) and
whole ecosystem warming (WEW or air warming) since their respective inception in all
treatment plots. (**B**) Percent of time belowground warming has been achieved since DPH targets
were achieved in 2014. (**C**) Percent of time air warming has been achieved since August of 2015.
NA = not applicable. All data are derived from daily mean air or soil temperature data.

| Treatment Target Temperature | +0 °C* | +2.25 °C | | +4.5 °C | | +6.75 °C | | +9 °C | |
|---|---|---|---|---|---|---|---|---|---|
| Plot # | 19 | 11 | 20 | 4 | 13 | 8 | 16 | 10 | 17 |
| A. Active Temperature Management | | | | | | | | | |
| DPH since 7/2/2014 (% days) | NA | 93.0 | 98.3 | 98.3 | 98.3 | 99.7 | 98.1 | 96.6 | 98.3 |
| WEW since 8/13/2015 (% days) | NA | 96.5 | 99.6 | 100 | 99.6 | 99.1 | 100 | 100 | 100 |
| B. DPH Statistics % Days within target °C | | | | | | | | | |
| Within 1.5 °C | 100 | 100 | 100 | 100 | 100 | 100 | 100 | 100 | 100 |
| Within 1.0 °C | 67.4 | 100 | 100 | 100 | 100 | 100 | 100 | 100 | 100 |
| Within 0.5 °C | 22.8 | 93.2 | 100 | 99.6 | 100 | 100 | 98.5 | 92.2 | 100 |
| Within 0.2 °C | 1.0 | 80.3 | 79.6 | 54.1 | 98.7 | 89.6 | 64.5 | 54.9 | 56.3 |
| C. WEW Statistics % Days within target °C | | | | | | | | | |
| Within 1.5 °C | 99.5 | 95.6 | 99.5 | 98.7 | 97.4 | 91.7 | 98.7 | 93.9 | 95.2 |
| Within 1.0 °C | 99.5 | 93.8 | 97.8 | 98.2 | 95.2 | 84.6 | 96.9 | 78.5 | 72.4 |
| Within 0.5 °C | 51.3 | 91.2 | 85.1 | 89.5 | 71.9 | 57.0 | 67.5 | 46.1 | 37.3 |
| Within 0.2 °C | 4.4 | 73.7 | 47.4 | 49.6 | 36.8 | 21.9 | 33.8 | 21.9 | 17.1 |

*Data for Plot #19 (the second constructed control plot with Plot 6 being the primary reference
for this table) reflect spatial variation rather than heating system performance.





Detailed plot-by-plot measured temperature data for both below and aboveground heating are
available for viewing at the following web portal http://sprucedata.ornl.gov.

A. Deep Peat Heating Profile

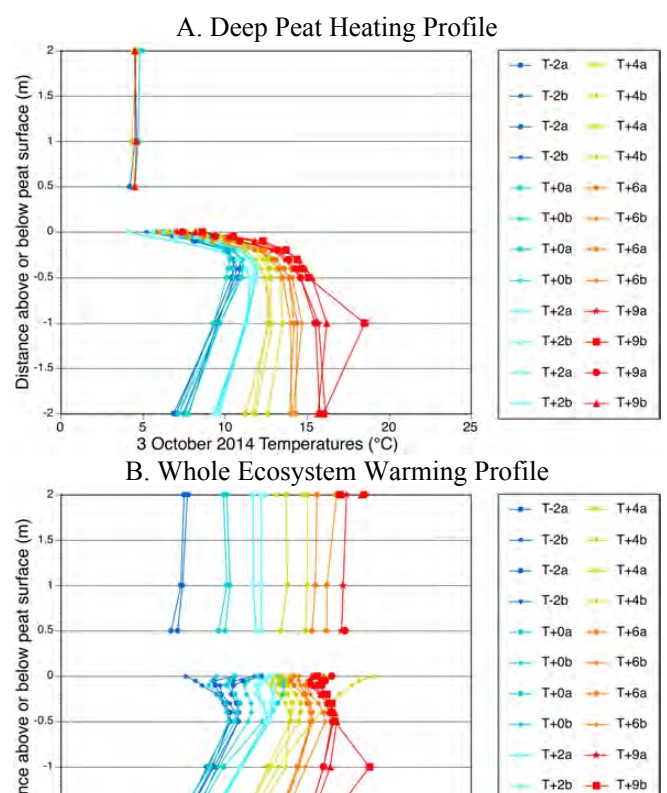


B. Whole Ecosystem Warming Profile


**Figure 5:** Temperature profiles from -2 m above through -2 m below the peat bog hollow surface
for (A) 3 October 2014 during deep peat heating, and (B) 3 October 2015 under whole
ecosystem warming. Air temperatures are the daily mean, and soil temperatures are the value
recorded at noon. Colors in the figure legend show data for unchambered ambient (T-2x), no-
energy-added control (T+0x) and warmed plots: +2.25(T+2x), +4.5(T+4x), +6.75(T+6x) and
+9(T+9x) °C, where x is either the a or the b series temperature zone within the plots.

**3.2 Temperature profiles within the enclosures**
During the period of DPH, and continuing under WEW, DPH in the -1 to -2 m peat depth was
achieved (Fig. 3). During DPH, air temperatures were not different, and surface peat
temperatures did not achieve the full target warming temperatures due to heat losses to the



atmosphere (Fig. 5a). With the addition of air warming, target temperature differentials were
approximated from the tops of the enclosed trees to peat depths of at least -2 m (Fig. 5b). The
data in Fig. 5 are only single snapshots of these type of data, and some variation over time in the
near surface peat zone is expected due to rain and snow events that may temporarily upset local
energy balance. The divergence of one peat temperature pattern in the B-series for one of the
+4.5 °C temperature plots (Fig. 5B) resulted from proximal heating of that particular zone of soil
by a heated air sampling tubing bundle. The heated bundle has since been repositioned to
eliminate this local bias.

Horizontal air temperature patterns are minimal within the plots due to the stirring of the internal
air by the fans of the air heating system and the coupling with external air exchanges (Fig. 2A).
These phenomena are fully described in the description of the prototype enclosure published
previously (Barbier et al. 2012), but color infrared temperatures provide quantitative data in
support of the distribution of horizontal temperatures within the plots (Fig. 6 and supplemental
data Fig. S4).

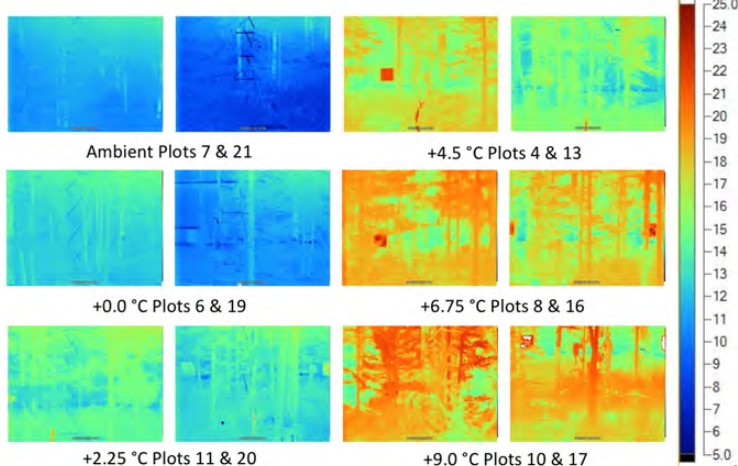

**Figure 6:** Color infrared images for the space within the designated treatment enclosures taken
on September 10, 2015 after sunset within a 30-minute period. The thermal color scale in °C
applies to all images. Non-biological metal or plastic surfaces in the images may not provide an
accurate temperature due to their emissivity difference from biological surfaces.





**3.3 Temporal Temperature variation**

It is useful to understand how both short (minute-by-minute) and longer-term (i.e., diurnal and seasonal) temporal variation within the enclosures compares between unchambered ambient and treatment plots. Figure 7 shows that control plots compare well to unchambered ambient conditions with almost no change in the standard deviation metrics for minute-by-minute observations within half hourly data. Conversely, the mean temperature standard deviations among one-minute data increase gradually with temperature treatments to nearly 2 times the nominal unchambered ambient standard deviation for the + 9 °C treatment plots. Increased short-term variance results from temperature control inefficiencies.

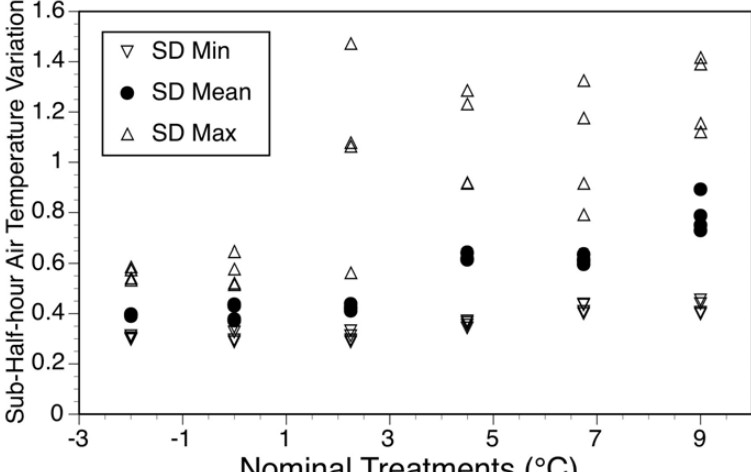

**Figure 7:** Sub-half hour variation of air temperature data expressed as the standard deviation of 1-min observations within a half hour measurement period. Plotted data are the daily minimum, mean and maximum half-hour temperature standard deviations over the period of observation for two replicate sensors in each treatment enclosure or plot. The -2 and 0 °C treatments in this graph represent unchambered ambient and no-energy-added control enclosures respectively.

Air temperature diurnal amplitudes for unchambered ambient plots ranged from 13.7 to 14.1 °C for warm season periods and 8.5 to 8.9 °C for cold season periods (Table 2). All treatment plot air temperature amplitudes remain within these diurnal ranges. Similarly, the unchambered ambient diurnal range for − 2 m soil temperatures lies between 0 and 0.2 °C, which is matched in the treatment plots.



**Table 2.** Range of diurnal air temperature amplitudes (AT, °C) at + 2 m in warm (DOY 230 to 300) and cold (DOY 300 to 365; 1 to 13) seasons, and the mean diurnal soil temperature amplitude (ST, °C) at – 2 m for a period including the warmest and coldest extremes of the measurement period (August 2015 – January 2016).

| Treatment and Plots | Ambient Plots (7,21) | +0 °C Plots (6, 19) | +2.25 °C Plots (11, 20) | +4.5 °C Plots (4, 13) | +6.75 °C Plots (8, 16) | +9 °C Plots (10, 17) |
|---|---|---|---|---|---|---|
| Warm season AT diurnal amplitude | 13.7 - 14.1 | 14.0 -14.1 | 13.0 - 13.7 | 13.3 - 13.5 | 13.9 - 14.2 | 13.2 - 13.6 |
| Cold season AT diurnal amplitude | 8.5 - 8.9 | 8.1 - 8.4 | 7.9 - 8.3 | 8.3 - 8.4 | 8.5 - 8.8 | 8.8 - 8.9 |
| -2 m soil temperate diurnal amplitude | 0.0 – 0.2 | 0.0 – 0.3 | 0.0 | 0.1 – 0.1 | 0.1 – 0.1 | 0.0 – 0.1 |

Annual amplitudes (approximated from summer maximums in 2015 and winter minimums in 2016) for air temperatures (49 to 51 °C) and soil temperatures at – 2 m (DPH: 4 to 5 °C; WEW 2.5 to 3.1 °C) are consistent among unchambered ambient and treatment plots (Table 3). The WEW system is capable of adding differential treatments to existing diurnal and seasonal temperature patterns.

**Table 3.** Annual range of observed maximum minus minimum air temperature at + 2m (AT, °C) for the whole ecosystem warming (WEW) period from August 2015 through January 2016, which includes the warmest and coldest periods of an annual cycle. Also shown are the range of maximum minus minimum soil temperatures (ST) at -2 m throughout the deep peat heating period in 2014 and 2015, and the WEW period since August 2015.

| Treatment and Plots | Ambient Plots (7,21) | +0 °C Plots (6, 19) | +2.25 °C Plots (11, 20) | +4.5 °C Plots (4, 13) | +6.75 °C Plots (8, 16) | +9 °C Plots (10, 17) |
|---|---|---|---|---|---|---|
| + 2 m AT for WEW | 50.4 - 51.1 | 50.2 - 50.5 | 50.5 | 50.2 - 50.5 | 50.6 - 50.8 | 49.1 - 50.5 |
| -2 m ST annual amplitude for DPH | 4.0 – 4.4 | 4.0 – 4.9 | 4.5 – 5.1 | 4.9 – 4.9 | 4.9 – 5.0 | 4.6 – 4.9 |
| -2 m ST annual amplitude for WEW | 2.4 – 2.5 | 2.6 – 3.1 | 2.6 – 2.8 | 2.9 – 2.9 | 3.0 – 3.0 | 2.6 – 2.9 |

### 3.4 Unchambered Ambient vs. Enclosure Environments

The mild belowground warming applied in SPRUCE produces minimal artifacts due to the deep soil target warming location and the low-wattage-heater application of energy. In contrast, the construction of walled enclosures to make air warming tenable produces a number of changes



from ambient conditions that need to be considered including: light, wind, humidity,
precipitation, dew formation, and snow and ice accumulation.

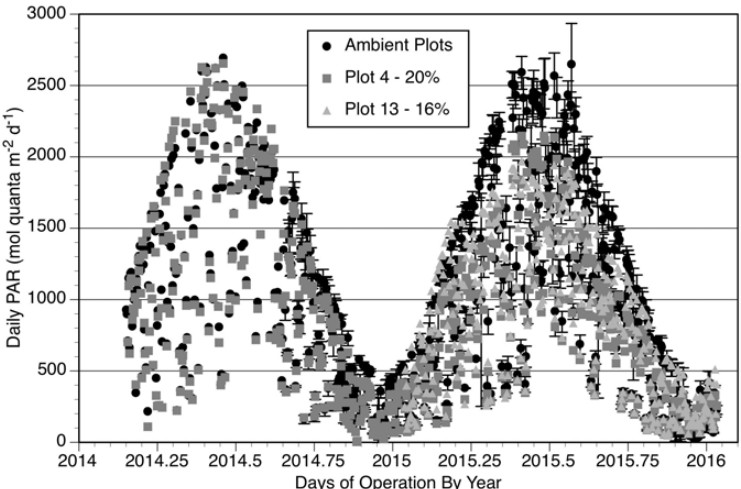

**Figure 8:** Example plot center daily photosynthetically active radiation (PAR) at 2.5 meters
above the bog surface in 2014 before enclosures were installed and after enclosure additions in
2015. The unchambered ambient plot data are from plot 7 (early in 2014) or the mean of plots 5,
7, and 21 with standard deviations shown. The figure legend shows the percent reduction in
annual cumulative PAR associated with the presence of the enclosure infrastructure.
Light levels within the plots before and after the installation of enclosures are plotted for selected
plots in Fig. 8. With the installation of the enclosure aluminum structure and the addition of
double-walled greenhouse glazing, midday PAR levels within the enclosures are reduced by
about 20 %. Under cloudy conditions or in the morning and evening when a greater fraction of
the light is diffuse, these differences are smaller. The greenhouse panels were not UV
transparent, but forest vegetation is known to largely tolerate UV light (Qi et al. 2010).

Short-wave and long-wave incident radiation data for the SPRUCE enclosures are reduced and
enhanced, respectively, when compared directly to matched data for unchambered ambient
conditions. Figure 9 shows examples of such data for a north and south centered location within
Plot 6 in the summer of 2016. When averaged over multiple mid-summer days the mean daily
reduction of incident short-wave radiation was 24.2±2.4 % at north plot locations and 40.9 ± 3.7
% for fully impacted southern locations (i.e., area of the plot subjected to all frustum, glazing and
wall frame influences). Opposite the effect for short-wave radiation, increases in long-wave



radiation incident on the surface showed a mean daily increase of 10 ± 2 % increase, but
increases were greater in the daytime than for nighttime conditions (Fig. 9).

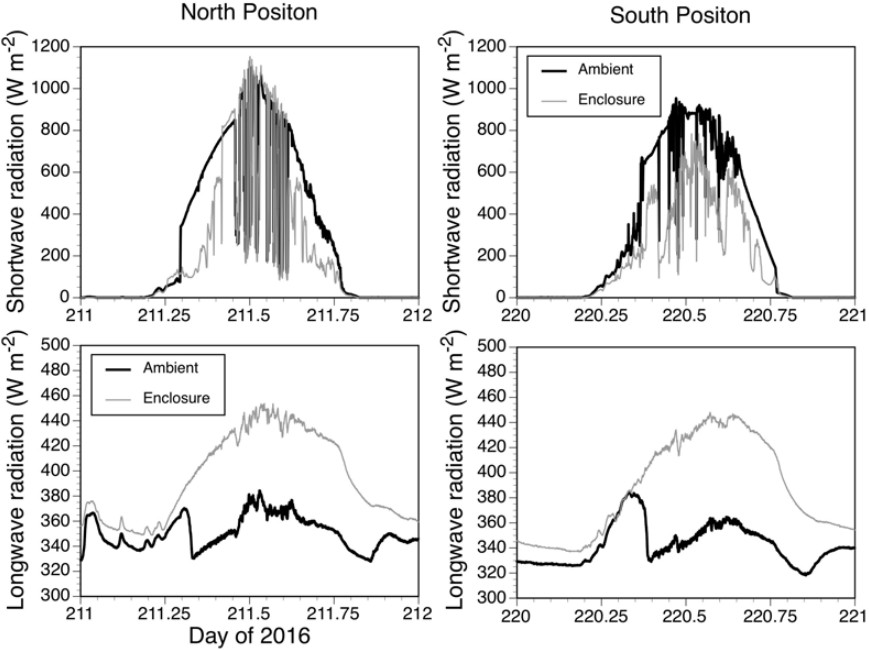

**Figure 9:** Example 1-minute incident short (upper graphs) and long-wave (lower graphs)
radiation data at north and south positions within the Plot 6 enclosure plotted against similar data
collected in unchambered ambient conditions. All data were collected approximately 2-m above
the surface of the S1-Bog boardwalks.

Ground level winds within the enclosures were necessarily enhanced to distribute heated air from
the edge sources to the center of the plot (Fig. 2A). To account for this enhanced wind effect, the
fully-constructed control applies the same air blowing system. While this provides a difference
between ambient conditions and treatment plots, it is fully controlled and comparable across all
heated enclosures. The air dynamics induced by external winds entering each enclosure through
the open top combined with internal turbulence generated by the blowers, homogenizes the air
volume inside the enclosure. Figure 10 shows a time series of vertical wind velocity and average
horizontal wind speed data contrasting unchambered ambient plots (Plots 2 and 21) with an
unheated enclosure (Plot 6) and the two +9 °C enclosures (Plots 10 and 17). There is more
turbulence in the enclosures than in ambient air and the turbulence increases with the level of
warming. Horizontal wind speeds are diurnally variable and comparable in both enclosed and



unchambered ambient plots. Vertical wind speeds are greater in the warming enclosures, increase
with level of warming, and are always in the upwards direction both day and night.

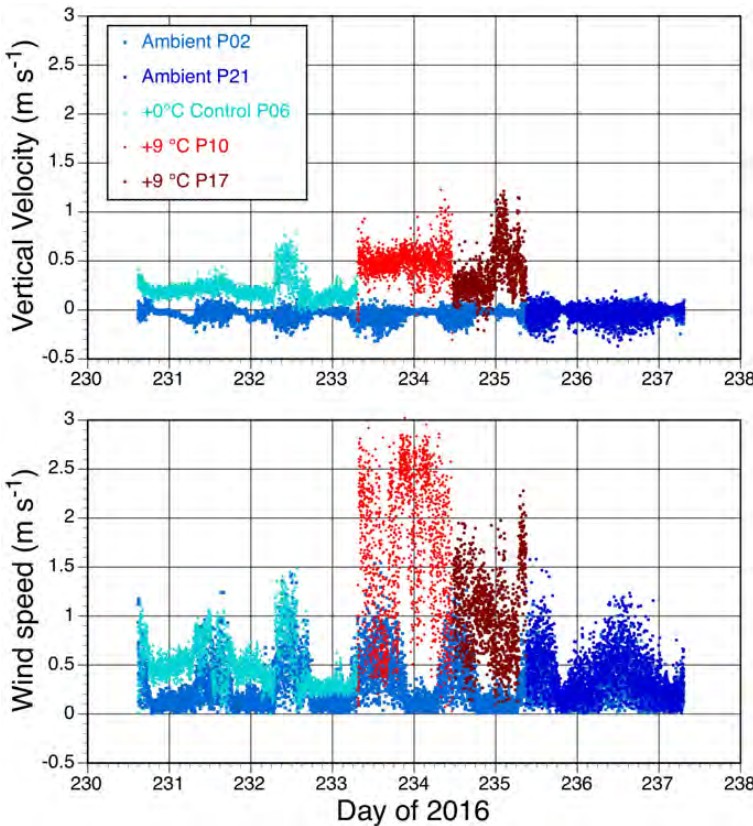

**Figure 10:** One-minute vertical wind velocity (Uz; upper graph) and mean horizontal wind
speed (Ux and Uy; lower graph) for unchambered ambient and enclosed plots of the SPRUCE
study during the summer of 2016.
Within the WEW enclosure total air turnover rates vary with external winds, and have been
measured using the dilution of constant $CO_2$ additions. At external wind velocities less than 0.5
m s$^{-1}$ the enclosure air turns over approximately one time every 5 minutes. As winds approach 8
m s$^{-1}$, the total air volume is turned over once per minute.

Absolute humidity within the enclosures is conserved across treatments (Fig. S5). This is
possible because of the wind induced turnover of air within the enclosures. Conversely, relative
humidity (Table 4) varies by treatment. The environment within the fully constructed controls



closely matches ambient relative humidity, but relative humidity within the warmed plots drops
proportionate to the warming treatments being only 51 to 55 % of the control for the most
extreme warming treatment (+ 9°C; Table 4).

Although common in ambient settings, dew formation has not been observed in any of the
warmed treatment enclosures, as relative humidity never reaches 100%. While this was to be
expected for the warmed plots, we were not certain if dew would form in the no-energy-added
control enclosures. In the control plots, RH does reach 100% on occasion, which would indicate
some dew formation. Even so, the foliage in the control plots has not been visibly wet in the
mornings, in stark contrast to the often heavy dew formation on foliage in unchambered ambient
plots.

**Table 4.** Plot-to-plot variation in mean daily relative humidity ±SD (RH; %) at +2 meters before
the construction of enclosures (A), with enclosures (B), with active air warming treatments
engaged during warm periods (C), and with heating during winter (D).

|  | Ambient Plots (7,21) | +0 °C Plots (6, 19) | +2.25 °C Plots (11, 20) | +4.5 °C Plots (4, 13) | +6.75 °C Plots (8, 16) | +9 °C Plots (10, 17) |
|---|---|---|---|---|---|---|
| A. Before* |  |  |  |  |  |  |
| Max RH | 99.0±0.2 | 98.8±0.0 | NA | 99.0±0.1 | NA | NA |
| Mean RH | 79.7±0.3 | 82.5±0.2 | NA | 79.3±0.1 | NA | NA |
| Min RH | 52.3±0.4 | 57.9±0.2 | NA | 52.6±0.0 | NA | NA |
| B. With Enclosures** |  |  |  |  |  |  |
| Max RH | 99.6±0.1 | 99.7±0.1 | 99.2±0.3 | 99.7±0.1 | 99.5±0.2 | 99.4±0.4 |
| Mean RH | 77.4±0.7 | 77.9±0.6 | 76.9±0.3 | 77.6±0.5 | 77.1±0.6 | 76.8±0.7 |
| Min RH | 48.7±0.9 | 50.1±0.5 | 49.2±0.3 | 49.7±0.6 | 49.4±0.4 | 48.9±0.2 |
| C. With Heating*** |  |  |  |  |  |  |
| Max RH | 99.4±0.3 | 96.7±0.5 | 83.8±1.8 | 76.7±2.4 | 66.0±0.5 | 58.8±0.7 |
| Mean RH | 81.8±1.0 | 78.1±0.2 | 66.3±1.5 | 60.1±1.8 | 51.1±0.1 | 45.1±0.5 |
| Min RH | 54.5±0.9 | 51.9±0.1 | 44.7±1.0 | 40.6±1.2 | 33.7±0.5 | 30.4±0.6 |
| D. Winter Heating**** |  |  |  |  |  |  |
| Max RH | 95.7±0.4 | 92.6±0.7 | 77.6±1.0 | 68.6±1.4 | 59.6±1.2 | 53.0±1.6 |
| Mean RH | 89.2±0.6 | 85.7±0.4 | 70.2±0.9 | 61.1±1.1 | 53.0±0.9 | 46.8±2.9 |
| Min RH | 77.0±0.4 | 73.1±0.3 | 58.8±0.6 | 50.0±0.5 | 43.9±0.7 | 39.3±4.1 |

*Days compared = days of the year 160 to 200 in 2014. ** Days compared = days of the year
160 to 200 in 2015; ***Days compared = days of the year 230 to 300 in 2015. ****Days
compared = days of the year 335 in 2015 to 10 in 2016. NA = not available.






Apparent water content and rate of soil drying also varies across plots due to the heterogeneous
density of hollows and differential tree density. Even so, the rate of soil drying increased when
the plot heating began, and drying was positively correlated with increasing plot temperatures
indicating enhanced evapotranspirational demand (Jeff Warren, personal communication).

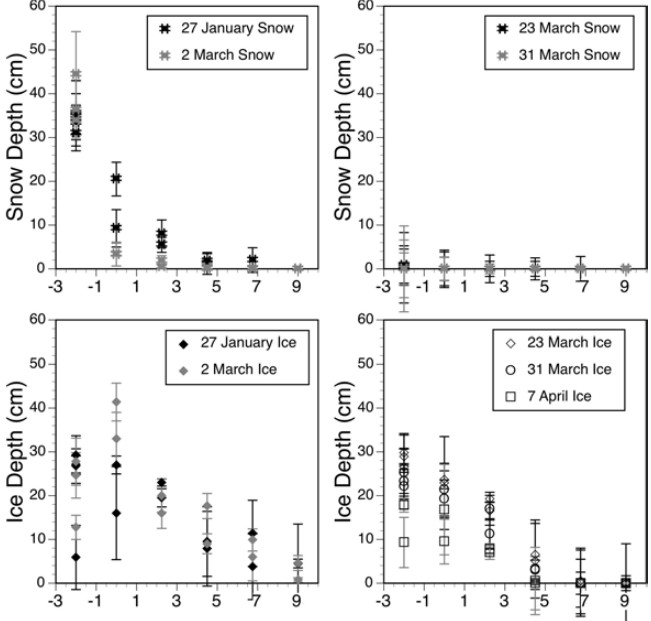

**Figure 11:** Snow depth (upper graphs) and ice depth (lower graphs) in each plot on January 27
and March 2, 23, 31 and April 7, 2016. All values are the mean depth ± sd for 4 locations within
replicate plots represented by the target treatment temperature differentials.
**3.5 Snow and Ice Accumulation**
An area of uncertainty in the development of the WEW prototypes in eastern Tennessee (Barbier
et al. 2012) was how snow accumulation would develop within the plots when deployed in
Minnesota. Observations throughout the winter of 2015-2016 have shown that snow actively
accumulates within the enclosures with a more or less uniform distribution around the plots (Fig.
S6). Ground level blower effects are limited to the edges of the plots (data not shown). Active
snow enters all warmed treatment plots, but its accumulation as a snow layer depends on the
temperatures of the vegetation and peat surface. Snow has been seen to accumulate in all warmed
plots if overall conditions allow, but it thaws or sublimates much faster in the warmed plots. The



control enclosures did not accumulate as much snow as ambient locations, but ice accumulation
within the peat profile can be equal to or greater than the accumulation in ambient areas at times
(Fig. 11). During the spring of 2016 the warmed plots lost their snow cover and ice thawed faster
than in the colder plots consistent with expectations for the experimental design.

**3.6 Energy Use**
The *in situ* WEW facility for tall statured plants was expensive to build yet cost-effective to
operate given the nature of the treatments. Key daily energy requirements for each treatment plot
under warm and cold season conditions are presented in Table 5. Soil warming using resistance

**Table 5.** Daily energy requirements for air and soil warming for the overall experiment and
values for individual treatment plots.

| Season | Warm Season Months (April to October) | | | Winter Months (November to March) | | |
|---|---|---|---|---|---|---|
| Treatment Energy Use | kW h d-1 | Gallons LPG d-1 | MJoules d$^{-1}$ | kW h d$^{-1}$ | Gallons LPG d$^{-1}$ | MJoules d$^{-1}$ |
| Air warming* | | | | | | |
|   Full Experiment | --- | 638 | 64,283 | --- | 795 | 80,102 |
|   By Treatment** | | | | | | |
| +0 °C Enclosure | --- | 0 | 0 | --- | 0 | 0 |
| +2.25 °C Enclosure | --- | ~31.9 | ~3,214 | --- | ~39.7 | ~4,000 |
| +4.5 °C Enclosure | --- | ~63.8 | ~6,428 | --- | ~79.5 | ~8,010 |
| +6.75 °C Enclosure | --- | ~95.7 | ~9,642 | --- | ~119.25 | ~12,015 |
| +9 °C Enclosure | --- | ~127.6 | ~12,857 | --- | ~159 | ~16,020 |
| Soil warming*** | | | | | | |
|   Full Experiment | 265 | --- | 954 | 495 | --- | 1,782 |
|   By Treatment | | | | | | |
| +0 °C Enclosure | 0 | --- | 0 | 0 | --- | 0 |
| +2.25 °C Enclosure | 9.0±1.7 | --- | 32.4±6.1 | 12.6±0.8 | --- | 45.4±3.0 |
| +4.5 °C Enclosure | 24.6±0.3 | --- | 88.6±1.0 | 31.9±2.9 | --- | 115.0±10.4 |
| +6.75 °C Enclosure | 38.8±7.1 | --- | 139.7±25.5 | 46.7±11.0 | --- | 168.3±39.5 |
| +9 °C Enclosure | 62.2±27.3 | --- | 223.9±98.2 | 69.4±21.2 | --- | 249.8±76.4 |
| Blower Energy**** | ~2,222 | --- | 7,999 | ~2,276 | --- | 8,194 |

*1 Gallon liquid petroleum gas (LPG US) = 100.757 MJ. **Air warming requirements by
treatment plots are only approximate and a derivation of total LPG use for the complete
experiment. ***Soil warming is measured by treatment plot, but is compared to metered energy
use by transect, which includes the energy for blowing air and the operation of instruments. 1
kW h = 3.6 MJ. ****Derived from total energy use during whole ecosystem warming minus
energy during deep peat heating for the respective periods.



heating was continuously measured in amps converted to kW h. Air warming using liquid
propane gas (LPG) for the full experimental site was estimated for each treatment in gallons of
LPG. Both energy units were converted to MJoules to make direct comparisons among the
warming methods. Air warming required 88 to 89% of the energy for WEW ranging from 64283
MJ d$^{-1}$ during the warm season to 80102 MJ d$^{-1}$ during cold months. Soil warming required only
1.3 to 1.9 % of the energy used ranging from 954 to 1782 MJ d$^{-1}$ of energy in the warm and cold
seasons, respectively. Although not a direct energy requirement for warming, 9 to 11 % of the
energy used was needed to drive the forced air blowers necessary to distributed warm air across
the 12 m diameter enclosures.

**3.7 Elevated CO$_2$ Treatments**
The capacity for adding pure CO$_2$ of known isotopic signature (obtained from an ammonia
production plant) to the air handling units of an enclosure to increase the atmospheric [CO$_2$] is
demonstrated in Fig. 12. Based on 6-min running mean observations we have sustained a + 500
ppm treatment within ±100 ppm using the current algorithms for a wide range of external wind
speeds (Fig. 12).

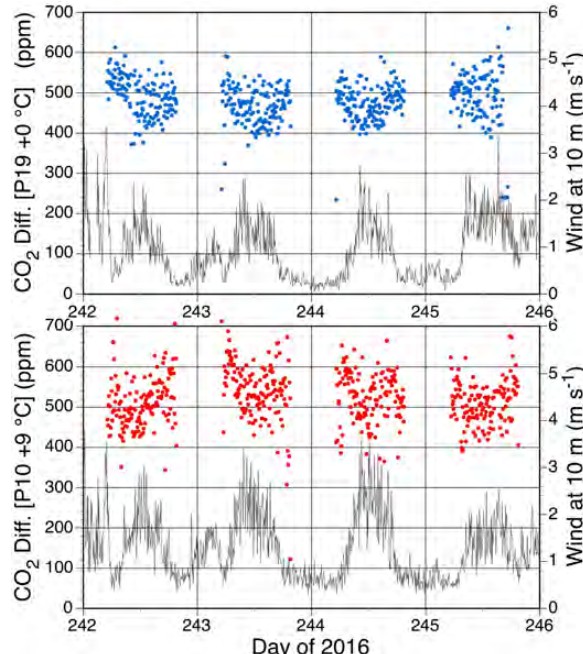

**Figure 12:** Examples of the differential CO$_2$ concentrations achieved over 4 days in 2016 for a
constructed control plot (+0 °C; upper graph) and plot warmed to +9 °C. All point data are 6-min



running mean [$CO_2$] differentials plotted with their respective 6-min running mean 10-m wind
speed data.

We are continuing to look at our control methods and will attempt to reduce the variation around
the target differentials. A comparison of these $eCO_2$ data with plot-to-plot variation for the non-
$eCO_2$ enclosures (Supplemental Table S5) suggests that the variation stems in part from spatial
variation hypothesized to be driven by localized differential air exchange between outside air and
the large enclosure volume. Warming and the buoyancy that it induces can also confound our
capacity to achieve a consistent +500 ppm $eCO_2$ treatment. The mean isotopic signature of the
elevated air was measured during the summer of 2016 as -22.6 ‰ $\partial^{13}C$ and -517 to -564 ‰ $\Delta^{14}C$.

**4.  Discussion**
Although there has been considerable discussion of the utility and merits of various warming
methods in recent years (Aronson and McNulty 2009; Amthor et al. 2010; Kimball 2011) we
chose to use air warming and deep soil warming for our studies, and have found the method
appropriate for warming a tall stature ecosystem (3 to 7 m) with active root and microbial
populations (> -2 m). The SPRUCE WEW enclosures provide us with the means to glimpse
warming futures at scales appropriate for the evaluation of peatland vegetation, microorganisms
and ecosystem functions. The SPRUCE enclosures are able to maintain the full range of
warming treatments (+2.25, +4.5, +6.75 and + 9 °C) over external wind velocities ranging from 0
to as much as 6 m s$^{-1}$. The system allowed the application of the warming treatments largely
uninterrupted throughout a full annual cycle. The experimental systems were successfully
installed in a sensitive wetland ecosystem with minimal visible impact on the target plot
vegetation and underlying peat column. The warming treatments provide a reasonable
approximation of projected future climate and atmospheric boundary conditions within which to
study a full range of vegetation, microbial and biogeochemical cycling responses.

**4.1 Comparing WEW to other methods**
Other notable studies using either air warming or direct surface warming via infrared lamps have
also been deployed to understand warming responses for a range of ecosystems (Table 6;
Aronson and McNulty 2009, LeCain et al. 2015, Rustad et al. 2001). Air warming methods for





Table 6. Comparison of the SPRUCE WEW system characteristics to other representative plot scale warming approaches operated in
field settings. Data are summarized at the individual plot level. Other warming studies not covered in this table of like studies are
summarized by Rich et al. (2015), Aronson and McNulty (2009), LeCain et al. (2015) and Rustad et al. (2001).

| Study/PI | SPRUCE WEW This Study | Black Spruce Plantation Bronson et al. 2008, 2009 | B4Warmed Rich et al. 2015 | PHACE LeCain et al. 2015 | Peatland Bridgham et al. 1999 | Temperate Seedlings Norby et al. 1997 |
|---|---|---|---|---|---|---|
| Ecosystem | Picea-Sphagnum Bog | Picea mariana plantation | Deciduous forest Understory with planted seedlings | Northern mixed prairie | Bog and Fen Monoliths | Old Field Chambers |
| Lat. / Long. (degrees) | 47.508 N -93.453 W | 55.883 N -98.333 W | 46.679 N; -92.520W & 47.946 N; -91.758 W | 41.183 N; -104.900 W | 47N; -92W | 35.903 N; -84.339 |
| Years of Operation | 2015 to 2025 | 2004 - 2006 | 2009 - 2011 | 2006 – 2013 (detail 2010-2013) | 1994 | Various Studies 1994-2004 |
| Differential treatments (+°C) | 0*, 2.25, 4.5, 6.75, 9 | 0*, 5 | 0*, 1.8, 3.5 | 0*, 1.5 Day/3.0 Night | 0*, 1.6-4.1 | 0*, 3 |
| Heated plot Area (m²) | 115.8 | 41.8 | 7.1 | 8.6 | 2.1 | 7.1 |
| Use of a constructed control | Yes | Yes | Yes | Yes | NA | Yes |
| Season and Diurnal Operation | 365 days, 24 hour | Heating treatments applied when control air > 0 °C | Warm season > 1 °C (208 to 244 days y⁻¹); 24 hour | 365 days, 24 hour | 365 days, 24 hour | 365 days, 24 hour |
| Aboveground Warming Method | Heated Air | Heated Air | Infrared Lamps | Infrared Lamps | NA | Heated/Cooled Air |
| Air T method and heights | Thermistors at 0.5, 1, 2(x2), and 4 m | Thermocouples at 1 and 2.5 m | IR Thermometer for the canopy surface | IR radiometers for the canopy/soil surface; Thermocouples at +25 cm, +15 cm (x2 within canopy) | NA | Thermistor 1 m |
| Volume of Heated Air surrounding vegetation (m³) | ~911 | ~209 | Not assessed | Not achieved | NA | 17 |



| Belowground Heating Method | Resistance heaters at 300 cm depth in an optimized pattern | Buried cables at -20 cm, 30 cm spacing | Buried cables at -10 cm, 20 cm spacing | NA | IR Surface Warming | Air Heating transfer |
|---|---|---|---|---|---|---|
| Soil T measurements and Depths (cm) | Thermistors at 0, -5 -10, - 20, -30, -40, -50, -100, -200 at three locations in each plots | -2, -5, -10, -25, -50, -100 | Type T thermocouples at -10 and a Subset at -20, -30, -50, -75, -100 | -0.5 cm, -3 cm | Thermocouple at -15 cm | Thermistor -10 cm |
| Soil Temp Control Depth (cm) | -200 | -20 | -10 | NA | NA | NA |
| Full Warming of soils below 1 m | Achieved | NA | Partial warming | NA | NA | NA |
| Volume of Fully Heated Soil (m³) | 232 | NA | ~2.1 | NA | NA | NA |
| | | | | | | |
| eCO2 Treatment | +500 µmol mol⁻¹ | None | None | 600 µmol mol⁻¹ | None | +300 µmol mol⁻¹ |
| eCO2 Seasons of Operation | Growing season/daytime | NA | NA | Growing season, daytime | NA | Growing season, daytime |
| Other Details | Hydraulically isolated to 3 to 4 m using a sheet-pile corral | Irrigated, VPD control with mist addition | Trenched | Hydraulically isolated to -60 cm | Extracted Monoliths | Evaporative coolers |
| | | | | | | |
| # Plots Operated | 10 | 8 | 72 | 10 | 27 | 12 |
| Design | Temperature Regression | 2 heat x 2 irrigation, Randomized Complete Block | 2 site x 2 habitat x 3 Temperature factorial | 2 heat x 2 CO₂ Factorial | 2 peatland types (bog and fen)x 3 heat x 3 water table factorial | Various factorial designs |

*A differential treatment of 0 implies the inclusion of fully constructed controls. NA = not applicable





field applications were established by Norby et al. (1997) for application to tree seedling and
Old-field research. They achieved air warming of +3 °C within 7.1 m$^2$ plots with limited soil
warming through air to soil heat transfer. Bronson et al. (2008, 2009) built larger air warming
chambers (41.8 m$^2$) combined with soil warming cables to study an upland *Picea mariana*
plantation at +1.8 and +3.5 °C air warming and partial soil warming (i.e., near surface).

Infrared lamp warming studies have also been successfully used to study warming effects for
some time (Harte et al. 1995), and most recent field-scale infrared lamp studies have employed
designs based on Kimball et al. (2008). Notable for comparison to the SPRUCE peatland work
was the study by Bridgham et al. (1999) that used constant output infrared lamps to generate
seasonally realistic warming from +1.6 to + 4.1 °C in extracted peat monoliths. More recently
and for *in situ* work in prairie systems, LeCain et al. (2015) deployed infrared lamps over
hydraulically isolated plots achieving variable day/night canopy warming of +1.5/+3.0 °C,
respectively, and surface soil warming at 3 cm depth up to 3.8 °C. Rich et al. (2015) describe a
warming study targeting temperate seedling responses in an upland forest with a system using
infrared lamps and buried cables over trenched plots to warm vegetation canopy surfaces to +1.8
and +3.5 °C. They reported significant warming within the soil profile, but did not achieve full
deep soil warming consistent with their above ground temperature treatments. Notwithstanding
the lack of deep soil warming and unassessed air warming, the Rich et al. (2015) study is very
impressive encompassing two sites and a total of 72 treatment plots deployed in a factorial
design. Infrared heating designs for much larger plots than those used by these groups have also
been proposed (Kimball et al. 2011), and one such study is currently underway in a Puerto Rico
tropical forest understory using 4-m diameter plots (Tana Wood, personal communication;
Cavaleri et al. 2015). Where vegetation canopies are short in stature so as to receive reasonably
uniform heat from infrared lamps, the infrared method provides a viable field method for
gathering temperature response data for vegetation and surface soil organisms.

The Hanson et al. (2011) deep soil warming protocols modified for SPRUCE are also being
adopted in other recent ecosystem studies. Whole-soil and mesocosm warming experiments are
being conducted in mineral soil (Caitlin Hick Pries, personal communication), and a salt marsh
warming study using a modification of the deep soil heating approach has been initiated at the



Smithsonian Ecological Research Center in Maryland (Pat Megonigal, personal communication).
Another approach has been to focus on single tree enclosures, as demonstrated by Medhurst et al.
(2006) who used fully-enclosed, aboveground whole-tree air warming of individual *Picea abies*
trees (8.3 m$^2$ plots) maintained air at +2.8 to +5.6 °C, and included eCO$_2$ control. That system
has subsequently been deployed for *Eucalyptus* studies in Australia (Barton et al. 2010). The
Medhurst approach was not fully integrated with belowground warming and associated
processes, but it did allow continuous assessments of the carbon exchange of the enclosed
vegetation. Whole-enclosure carbon exchange calculations are planned for the SPRUCE study
using a modified eddy flux constrained assessment for ambient-CO$_2$ enclosures (Lianhong Gu,
personal communication).

Less technologically intense passive studies of warming, not covered in the reviews mentioned
earlier, include a peat monolith transplant study down an elevation gradient allowing the
characterization of a +5 °C temperature change (Bragazza et al. in press), a snow depth
manipulation deployed in the arctic (Natali et al. 2011), and evaluations of thermal gradients
around a geothermal source in Iceland (O'Gorman et al. 2015). While differing in plot sizes,
level of above and belowground temperature control or assessment, and the ability to standardize
methods, these approaches represent alternate methods from which to gather information on
vegetation and microbial system responses to warming.

**4.2 Unique Characteristics of the WEW Method**
The following text describes and discusses the influence of the WEW enclosures and treatments
on environmental variables that were altered from expected ambient conditions including: light,
wind, humidity, precipitation, ice and dew formation.

**4.2.1 Light**
The presence of greenhouse glazing and the enclosure structure reduced incident PAR at the
center of the enclosures by around 20% during midday periods. This level of reduction is not
sufficient to limit the photosynthetic capacity of the *Picea* foliage (Jensen et al. 2015) nor the
other photosynthetic forms of vegetation being studied (Jeff Warren, personal communication).
Reductions in short-wave radiation ranged from 24 to 41% and varied within the enclosure along



a south to north gradient. Long-wave or far infrared radiation representative of sky/cloud
temperature conditions were 10% greater than for ambient conditions leading to less heat loss at
night in constructed chambers when compared to unchambered ambient plots.

**4.2.2 Wind**
The increase in enclosure turbulence in warming and control plots is driven by forced air
movement from the hot air blower system, and confounded by the influence of vertical warm air
buoyancy. Increased horizontal turbulence is present in the unheated control enclosures
($0.14\pm0.24$ to $0.31\pm0.23$ m s$^{-1}$), and much larger in the +9 °C heated chambers ($0.8\pm0.4$ to
$1.3\pm0.9$ m s$^{-1}$). Vertical velocities ($U_z$) in the control and +9°C plots, show increases of
$0.26\pm0.18$ m s$^{-1}$ for the Plot 6 control, and for the ±9 °C treatment enclosures $0.55\pm0.14$ m s$^{-1}$ for
Plot 10 and $0.41\pm0.24$ m s$^{-1}$ for Plot 17. A more detailed analysis of turbulence patterns across
the full range of warming enclosures will be evaluated in the future with planned deployment of
eddy flux instrument packages within the ambient-$CO_2$ enclosures for whole-enclosure-footprint
$CO_2$ and $CH_4$ flux measurements.

**4.2.3 Atmospheric humidity**
Warming of the enclosure using air containing consistent absolute humidity (supplemental data
Fig. S5) led to proportionate reductions in relative humidity (Table 4) and sustained a higher
gradient of vapor pressure between the well mixed enclosure air and wetter soil and plant
surfaces. Although not to the levels induced by the SPRUCE treatments, the most recent IPCC
report (Collins et al. 2013) concluded that relative humidity over interior continental regions
could be projected to drop with future warming. Some prior warming studies have considered
how to ameliorate this drop in humidity and reduction in soil water use by use of a steam/misting
system or irrigation in warmed plots (e.g., Bronson et al. 2008, 2009; De Boeck (2012)

Steam addition to sustain relative humidity within small open-topped warming chambers has
been shown to be technologically feasible (Hanson et al. 2011), however, it was not considered
for deployment at SPRUCE due to the requisite energy costs and water volume requirements.
For example, let us assume a mid-summer condition (25 °C, 97 kPa, 90-100 % day/night RH)
and continuous operation of our 911 m$^3$ open top enclosures at + 9 °C with a mean external wind



velocity of 2 m s$^{-1}$, an enclosure turnover fraction of approximately 0.62 (actually external winds
and turnover fractions are often much greater), and a day/night RH of 47/70 %. Under these
conditions, a water source of 9.7 m$^3$ d$^{-1}$ would have been needed for routine operations along
with additional energy to convert it to steam would have been required to sustain the ambient
relative humidity of 90% within the +9 °C enclosure. Such a distilled water supply (necessary to
limit corrosion and nutrient transfers to the ecosystem) and energy supplies made RH control too
expensive. A mist based approach for controlling humidity in a free air environment has been
reported (Kupper et al. 2011), but such a system still requires the availability of a significant
treated water source and would increase the air warming heating demands necessary to sustain
our air warming differential temperatures due to the latent heat absorbed by evaporating droplets.

Choosing to operate our WEW system with variable relative humidity led to greater proportional
surface evaporation from *Sphagnum* (essentially all ground cover), water use by C3 plants and an
expected reduction in the seasonal water table with warming. In the first season of operation,
reductions in water table depths were limited as the corralled plots were left undrained and
ambient rainfall inputs exceeded losses from evapotranspiration. Since relative humidity was
allowed to vary with treatments in SPRUCE, significant effort was invested in fully quantifying
the impact on changing surface sphagnum and peat water content, plot level water balance, and
water table depth within each enclosure.

**4.2.4 Precipitation and Winter Ice**
Although the frustum encircling the top of the enclosure does create an internal rain and snow
shadow over the internal boardwalk, the excluded rain runs down the enclosure walls onto the
peat surface inside of the corral barrier. As a result, there is a rain shadow impact for some edge
vegetation, but the overall water inputs to the plot remain the same as for an unchambered
ambient plot (data not shown). The frustum does, however, reduce winter snow accumulation
within the plot because some snow is thrown clear of the subsurface corral (Fig. 11). However,
ice formation in the surface peat of the control plots was similar to or greater than that found
beneath unchambered ambient plots (Fig. 11).





Changes to the energy balance due to the presence of the enclosure (described above) have a
large impact on snow depth between unchambered ambient and enclosed plots. Simulations with
the CLM-SPRUCE model indicate that on average, the snow depth is reduced by 40% in
enclosed vs. unchambered ambient plots, with the highest reductions in the late winter and early
spring. Complete loss of snowpack generally occurs 2-3 weeks earlier when the effects of the
enclosure are considered. The observed reductions are slightly larger, reflecting enclosure snow
shadowing effects and potentially higher sublimation caused by increased air movement not
considered in the simulations. Despite the reduction in snow cover, the simulated ice depth is
similar between the unchambered ambient and enclosed plots – and this correlates well with our
*in situ* observations (Fig. 11). The warming of the peat layers caused by increased longwave
input is likely compensated to a large degree by increased heat loss during cold snaps because of
the reduction of insulating snowpack, an effect that was explained in more detail in Shi et al.

860    (2015).


**4.2.5 Lack of dew formation**
Even without active warming, modifications to the energy balance caused by the enclosures lead
to warming effects that influence air and vegetation temperatures, dew formation and snow
dynamics. The incoming longwave radiation within the enclosure is significantly elevated,
especially in clear-sky conditions. Simulations with the CLM-SPRUCE model (Shi et al., 2015)
were conducted to investigate the effects of SPRUCE enclosures on changes in the energy
balance on dew formation, snowpack and soil ice. Simulated average 2m air temperatures within
the enclosures are about 0.8 °C warmer than the unchambered ambient plots. This warming
effect is highly variable, ranging from nearly zero to over 5°C, and is largest in the early morning
under clear conditions, when radiation cooling is inhibited most by the enclosure walls, and
during the winter months when longwave radiation is a larger fraction of the overall radiation
budget. While the observed differences follow this general pattern, they are more than double the
simulated magnitudes. This may be due to the model ignoring the impacts of the enclosure on
wind speed and turbulence patterns, which cannot be considered in these simulations because the
assumptions in CLM-SPRUCE about Monin-Obukhov similarity and logarithmic wind profiles
(Oleson et al., 2013) that cannot easily be extended to the SPRUCE conditions. Simulated leaf



surface temperatures in the enclosures were elevated on average by 2.5C, which has important
implications for carbon and energy fluxes.

Despite the underestimate of air warming in the simulation, the model results indicated a near
complete inhibition of dew formation (Fig. 13), similar to site observations. Total dew

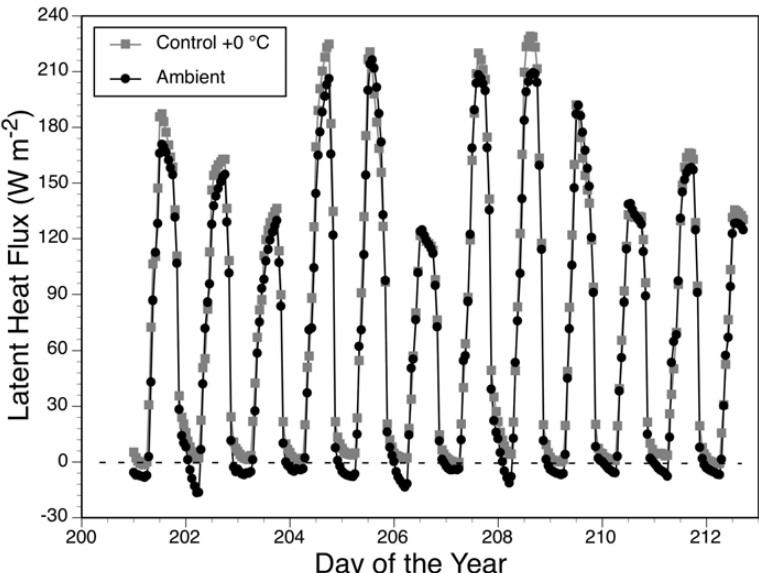

**Figure 13:** Simulations of latent heat flux over a 10-day period for ambient conditions (black)
and in a control enclosure (grey) using environmental driver meteorology data from July 2013.
Negative latent heat fluxes indicate dew formation, but only occur for the ambient condition.
formation was about 12mm integrated over the growing season (May-September) in the ambient
simulation, but only 0.5mm in the enclosure simulation (96% reduction). In the simulations, this
resulted from higher surface temperatures and lower relative humidity. Near-surface wind speeds
in the enclosures are also usually higher than for unchambered ambient areas as a result of the
blowers. This turbulence likely further inhibits the formation of dew, but such an effect was not
considered in the CLM simulations.

**5.  Conclusion**
The WEW system described is capable of providing a broad range of warming conditions up to +
9 °C with minimal artifacts from the experimental infrastructure. The end result is an experiment
system capable of giving scientists a fair glimpse of organism and ecosystem responses for





plausible future warming scenarios that can't be measured today or extracted from the historical
record. The large SPRUCE enclosures allow ongoing ecosystem-level assessments of warming
responses for vegetation growth and mortality, phenology changes, changing microbial
community composition and function, biogeochemical cycles and associated net greenhouse gas
emissions.

**6. Data Availability**
The environmental measurement data referenced in this paper are archived at and available from,
the SPRUCE long-term repository (Hanson et al. 2016; http://mnspruce.ornl.gov).

**7. Author Contributions**
P. Hanson conceived the experimental methods and wrote this paper. C. Barbier optimized the
air warming system using complex fluid dynamics models. J. Riggs programmed the SPRUCE
enclosure feedback control systems. M. Krassovski designed and maintained the local and
satellite communications systems. P. Hanson, W.R. Nettles, J. Phillips, J. Riggs and J. Warren
installed and maintain instrumentation. A. Richardson supplied installed and monitored plot
phenology cameras. D. Aubrecht evaluated light transmission characteristics of the enclosure
sheathing. L. Gu interpreted wind velocity and speed data. D. Ricciuto executed runs of the
CLM-SPRUCE model to interpret enclosure energy balance properties. LA Hook archived data.
All authors have read, understand and agree to the content of this paper.

**8. Acknowledgments**
The authors would like to thank Dr. Randall K. Kolka, USDA Forest Service for working in
collaboration with the Oak Ridge National Laboratory to enable access to and use of the S1-Bog
of the Marcell Experimental Forest for the SPRUCE experiment and affiliated studies. We would
also like to thank Natalie A. Griffiths, Stan D. Wullschleger and Randall K. Kolka for their
comments on early drafts, as well as editorial assistance provided by Terry Pfeiffer.

This material is based upon work supported by the U.S. Department of Energy, Office of
Science, Office of Biological and Environmental Research. Oak Ridge National Laboratory is
managed by UT-Battelle, LLC, for the U.S. Department of Energy under contract DE-AC05-



00OR22725. The development of PhenoCam IT infrastructure was supported by NSF's
Macrosystems Biology program (award EF-1065029).



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
