# Peer review of "Attaining Whole-Ecosystem Warming Using Air and Deep Soil Heating Methods with an Elevated CO2 Atmosphere"

_Biogeosciences, 2016_

## Referee Comment (RC1) · Anonymous Referee #1 · 4 Jan 2017

The following is a review of the manuscript "Attaining Whole-Ecosystem Warming Using Air and Deep Soil Heating Methods with an Elevated CO2 Atmosphere." This manuscript details a newly developed air and soil warming study with elevated CO2, located in the boreal forest of Northern Minnesota. The manuscript outlines the methods for achieving warming of soil and air, along with elevated CO2. Undoubtedly this will be the foundational methods paper cited in future research articles.

Scientific significance: These types of large warming+CO2 studies are highly valuable to the understanding of future climate scenarios and modeling of ecosystem carbon fluxes. This manuscript not only focuses on a study design that emphasizes temperature response functions, but tests a temperature increase much higher than past

boreal warming studies (+9 C), which sadly could be a realistic scenario that hasn't been thought possible in earlier boreal warming studies. This study has the potential to significantly improve the current understanding of how boreal systems respond to warming and elevated CO2, especially in respects to carbon budgets.

Scientific quality: The work that has gone into the outlined study is of high quality. The study design has been well thought out. The infrastructure to achieve the soil and air warming along with elevated CO2 has been well tested and this manuscript illustrates the ability of the authors to achieve the goals of the study.

Presentation quality: The manuscript is well written, easy to comprehend and illustrates two years of environmental manipulation. Below I pose a few questions along with a general comment for the authors and editor to consider.

Overall, I believe this manuscript to be worthy of publication in Biogeosciences.

General comment: Hydrologic responses: An important component that I think is lacking in this manuscript are data relating to hydrologic changes due to the experimental manipulation. The hydrologic conditions drive this ecosystem, limiting decomposition and nutrient availability, while also suppressing soil carbon fluxes. The authors have chosen to allow soil drying (a viable future scenario) to occur with warming in this study. Lines 634-637 state that soil drying was correlated with plot temperatures, which is what readers would expect. However, readers will be interested to know the rate of change and magnitude to the water table with the various warming treatments. I would think a figure illustrating water table fluctuations and differentials between treatments would be very important. If the authors can provide data for the readers, it would be greatly appreciated.

Specific comments: Lines 147-158: Could you state the number of trees per open top chamber/plot, maybe it is a range?

Line 183: Was the regeneration of the black spruce natural or artificial? Trees are 5-8

meters tall, but what is range in diameter? This will help readers better understand growth rates. I didn't see where the height of the chambers was mentioned. Please add this unless I missed it.

Figure 5 "Temperature profiles from -2m above through -2m below" : I have read this line a few times and I know what you are saying, but is the first -2m a typo? Did you mean to say 2m above the peat surface through -2 m below the peat surface? Something to look at.

---

## Referee Comment (RC2) · Anonymous Referee #2 · 12 Jan 2017

Thanks for the opportunity to review this paper. Overall it was very informative and is suitable for publication with some minor revisions. I believe that the authors do a good job informing the audience about the development and design of the colossal SPRUCE endeavor. This is no easy task and I think that the authors are 95% of the way there. I am somewhat less satisfied with the comparison with other approaches, as I do not think they have enough space to go as deeply as I would like. I will make a couple for suggestions for that section of the paper along with some comments related to the presentation of experimental results. My strongest concern about this paper is that the manner in which the data is presented does not let the reader really evaluate the effectiveness in context rooted to temporal ecological processes. They have effectively

shown how on average SPRUCE works. I would like to see the data presented in a slightly different manner that would also allow a deeper dive into understanding (from and ecosystem context) where the approach successful and limited. This would help readers with hypothesis development and aid the discussion limitations and successes. The experimental objectives are to replicate ambient conditions while altering only the change factors we have chosen at all spatial and temporal scales of the experiment. Thus, it is important to show experimental function in this manner. This would start by showing the distribution of above and belowground temperature data for each of the treatments. It is important to present at least some the data in a manner that does not just show that the treatments are different on average, for narrow bands of time. Rather I would like to see some exploration of the daily and annual patterns observed versus what we would expect to see. As this data is currently presented, there is strong difference in the daily averages of each treatment and they seem to be consistent throughout the year. But these data lump seasonal and diurnal variability and may mask patterns of efficacy that are important for the reader to understand. The authors should use the delta from ambient as a measure of the experiment look at the average and variability across various ecological scales. Hour of day (not just an individual day) would be the most important but also by time of year. The limited number of sensors makes spatial variability harder to explore in this manner but it would be important as well. I would also like to see the overall distribution of temperatures for each of the treatments. It is important that the distribution of temperatures match ambient as much as possible along with differing in mean. Some of the papers they reviewed in this ms use analyses like those suggested, I would also the see if there are seasonal patterns as well. It is easier to use the deltas for these analysis then the overall temperature. It is likely that variability in treatment is higher in parts of the day or times of the year and that would be important to know. I would also like to see multivariable traces and deltas for 10 days or at an hourly scale. This could be in the supplement and help the reader see the efficacy of the experiment in an ecological context. It is probably beyond the scope of this paper but I would like to see an analysis

linking directly the specific temperature/ light and rh conditions of sampling area with measurements just in thoses areas. I am not sure what spatial data is available but it would reassure readers to know that the sampled area variability is minimized. As it is a whole ecosystem model with some range in values, it would be nice to know whether the sampling area occupies that entire range or is experimenting a narrower range of treatments. For example, it would be great if RH decline with temperature in areas sampled was less than chamber level. I am especially concerned about pattern of nighttime temperature with distance from chamber wall and RH variation with distance from blower manifold. There is very little discussion of soil temperature behavior during freezing and thawing cycles or by depth. This need to be include somewhere. I expect that soil and air temperatures invert at some point during the year and it might be better analyze these data separately. Again a delta based analysis of soil temperature differences would be better to show treatment effects compared to ambients rather that overall temperatures. It would be nice to know that the delta variabilty at each depth was comparable with ambient. Daily pattern in RH would also be nice to know as well. 211- Is this really 12-18m deep below wetland. Please check. The figure sharpness seemed lacking throughout, I assume that will be corrected. I like figures with sd bars rather than separate symbols Table 2 explanation was confusing to me. Soil moisture data to back up discussion of RH and ET? Figure 7 is good. I would like to see more like this. I would like to see the same analyses for differing sensor variables. One could be in paper and other in supplement. Right now the comparison discussion between this and other warming experiment seems underdeveloped. I suggest picking a couple of key comparisons to develop discussion. Temporal pattern, dewpoint, soil moisture and RH be interesting to include more of. What should RH and dewpoint looked in a good manipulation? The summary table also need to be checked. The data from at least one of these papers is incorrect.
* * *

---

## Author Response (AR1)

**Point-by-point Response to Reviewers #1 & #2 for bg-2016-449**

**"Author Responses are indented and in bold-text, and specific changes in the manuscript are noted in blue text"**

RC1 – Anonymous Referee #1

The following is a review of the manuscript "Attaining Whole-Ecosystem Warming Using Air and Deep Soil Heating Methods with an Elevated CO$_2$ Atmosphere." This manuscript details a newly developed air and soil warming study with elevated CO$_2$, located in the boreal forest of Northern Minnesota. The manuscript outlines the methods for achieving warming of soil and air, along with elevated CO2. Undoubtedly this will be the foundational methods paper cited in future research articles.

> **No response required.**

Scientific significance: These types of large warming+CO2 studies are highly valuable to the understanding of future climate scenarios and modeling of ecosystem carbon fluxes. This manuscript not only focuses on a study design that emphasizes temperature response functions, but tests a temperature increase much higher than past boreal warming studies (+9 C), which sadly could be a realistic scenario that hasn't been thought possible in earlier boreal warming studies. This study has the potential to significantly improve the current understanding of how boreal systems respond to warming and elevated CO$_2$, especially in respects to carbon budgets.

> **We appreciate the supportive comments of Referee #1.**

Scientific quality: The work that has gone into the outlined study is of high quality. The study design has been well thought out. The infrastructure to achieve the soil and air warming along with elevated CO$_2$ has been well tested and this manuscript illustrates the ability of the authors to achieve the goals of the study.

> **Thank you for recognizing our effort. We have indeed attempted to produce a system that allows a fair glimpse of plausible future environments.**

Presentation quality: The manuscript is well written, easy to comprehend and illustrates two years of environmental manipulation. Below I pose a few questions along with a general comment for the authors and editor to consider. Overall, I believe this manuscript to be worthy of publication in Biogeosciences.

> **Thank you.**

General comment: Hydrologic responses: An important component that I think is lacking in this manuscript are data relating to hydrologic changes due to the experimental manipulation. The hydrologic conditions drive this ecosystem, limiting decomposition and nutrient availability, while also suppressing soil carbon fluxes. The authors have chosen to allow soil drying (a viable future scenario) to occur with warming in this study.

Lines 634-637 state that soil drying was correlated with plot temperatures, which is what readers would expect. However, readers will be interested to know the rate of change and magnitude to the water table with the various warming treatments. I would think a figure illustrating water table fluctuations and differentials between treatments would be very important. If the authors can provide data for the readers, it would be greatly appreciated.

**Strong drivers for changes in hydrologic response are certainly apparent from the warming induced changes in atmospheric relative humidity (Table 4), and we agree that potential drying under warming climate scenarios is a key variable of interest related to both microbial and vegetation responses. We are monitoring surface drying with capacitance probes (see Supplemental Material section) and overall plot water status with central water table depth sensors (where the zero height is defined as the mean hollow height for the peatland plot.**

**During the initial years of air warming operations 2015 and 2016, rainfall occurred in an abundance and at a frequency that did not allow sustained drying of the peatland for any treatment. Nevertheless, as noted by the reviewer, the enhanced drying potential with warming should be evident. This is most easily demonstrated in a cumulative manner through the accumulation of winter snow (Figure 11 showing less snow with warming). It is also evident in the dynamics of surface peat drying (on site observations), but is not as easily captured along the warming gradient by the capacitance sensors. A new figure showing mid-summer 2016 surface peat hollow moisture for contrasting the extremes of the warming treatments (control Plot 19 vs. the mean of hollow sensors in the +9 °C warmed plots 10 and 17) is provided below. Other plots fall between these values.**

**Two members of the SPRUCE team (Steven D. Sebestyen and Natalie A. Griffiths) are also actively engaged in the detailed monitoring and interpretation of the water table levels and plot-scale quantification of outflow quantities and chemistries. The methods for collecting such "response" data have been summarized in the following 26-page archived description.**

**Sebestyen, S.D., and N.A. Griffiths. 2016. SPRUCE Enclosure Corral and Sump System: Description, Operation, and Calibration. Climate Change Science Institute, Oak Ridge National Laboratory, U.S. Department of Energy, Oak Ridge, Tennessee, U.S.A. http://dx.doi.org/10.3334/CDIAC/spruce.030**

**The current manuscript has been revised to include this reference (Line 1226) and the text has been supplemented (Line 168, 920) to suggest that such data will be forthcoming in another article dedicated to hydrologic changes induced by the SPRUCE treatments.**

This new Figure S2 was added on page 55 to the Supplemental Material for the paper.

[Figure]

**Figure S2**.  Graph of half-hour rainfall at + 6 m (upper graph) and surface peat water content averaged over 0 to -10 cm (lower graph) during a mid-summer dry period during 2016.  Small precipitation events are intercepted by the canopy and peat *Sphagnum* surface and have limited effects on bulk water content observations.

Specific comments:

Lines 147-158: Could you state the number of trees per open top chamber/plot, maybe it is a range?

**All saplings greater than 1 cm diameter at 1.3 m above the Sphagnum surface are defined as trees for the SPRUCE study.  Within the interior boardwalk of each plot or enclosure the number of trees ranges from a minimum of 10 larger trees in Plot 10 to a maximum of 27 trees in Plot 20 for a mean number of trees per plot of between 18 and 19 whole trees.  This information has been added to a modified description of site vegetation within Section 2.2.**

Line 183: Was the regeneration of the black spruce natural or artificial? Trees are 5-8 meters tall, but what is range in diameter? This will help readers better understand growth rates. I didn't see where the height of the chambers was mentioned. Please add this unless I missed it.

**All regeneration following the strip cut events in 1969 AND 1974 occurred through natural vegetative processes or seeding events (3 to 4 successful events since 1969). Tree diameters at 1.3 m range from a plot mean minimums of 3.5 cm to plot mean maximum of 6.5 cm with a mean plot tree diameter of 5.2 ± 0.9 cm. The full range of dbh ranges from 1.2 to 11.1 cm. This information has also been added to a redrafted Section 2.2.**

Figure 5 "Temperature profiles from -2m above through -2m below": I have read this line a few times and I know what you are saying, but is the first -2m a typo? Did you mean to say 2m above the peat surface through -2 m below the peat surface? Something to look at.

**The Figure 5 legend text (new lines 490 to 495) was in error and has been corrected to state "2m above the peat surface through -2 m below the peat surface".**

Response to Reviewer #2

Thanks for the opportunity to review this paper. Overall it was very informative and is suitable for publication with some minor revisions. I believe that the authors do a good job informing the audience about the development and design of the colossal SPRUCE endeavor. This is no easy task and I think that the authors are 95% of the way there.

**Thank you. No response required.**

I am somewhat less satisfied with the comparison with other approaches, as I do not think they have enough space to go as deeply as I would like. I will make a couple for suggestions for that section of the paper along with some comments related to the presentation of experimental results. My strongest concern about this paper is that the manner in which the data is presented does not let the reader really evaluate the effectiveness in context rooted to temporal ecological processes. They have effectively shown how on average SPRUCE works. I would like to see the data presented in a slightly different manner that would also allow a deeper dive into understanding (from and ecosystem context) where the approach successful and limited. This would help readers with hypothesis development and aid the discussion limitations and successes.

**The full data sets on performance for half-hour, above and below ground temperature responses, and aboveground $CO_2$ levels are archived and available in Hanson et al. (2016). For the 12 experimental plots covered by this initial project data set there are already over 23,000,000 observations (by plot: wind x2, air temperature x5, soil temperature x33, relative humidity x5, rain x1, PAR x1), and over 21,000,000 assessments of variation within half-hour periods. In the paper, we summarized concisely the nature of the response data for these variables over short to long term time intervals.**

**Hanson, P.J., Riggs, J.S., Nettles, W.R., Krassovski, M.B., Hook, L.A.: *SPRUCE Whole Ecosystems Warming (WEW) Environmental Data Beginning August 2015*. Carbon Dioxide Information Analysis Center, Oak Ridge National Laboratory, U.S. Department of Energy, Oak Ridge, Tennessee, U.S.A. http://dx.doi.org/10.3334/CDIAC/spruce.032, 2016.**

**In the following text, we provide responses to Reviewer #2's questions and recommendations for the improvement of our paper.**

The experimental objectives are to replicate ambient conditions while altering only the change factors we have chosen at all spatial and temporal scales of the experiment. Thus, it is important to show experimental function in this manner. This would start by showing the distribution of above and belowground temperature data for each of the treatments.

**The objectives are to add temperature (or $CO_2$) differentials onto existing ambient patterns while conserving (as much as possible) the natural half-hour, diurnal, and seasonal patterns of the ambient environment. We have already attempted to illustrate this conservation for half-hour data in Figure 7, diurnal data in Table 2, and seasonal/annual amplitudes in Table 3. As we understand the reviewers new request for additional "distribution" data, we have constructed the following histograms for the half hour data set over the period of observations archived in Hanson et al. (2016).**

[Figure]

**New Figure 10:  Frequency distributions for daily mean soil temperature at -2 m (left column), air temperature at +2m (middle column), and daily mean relative humidity at + 2m (right column) throughout the evaluation period in 2015 and 2016.  Data in the frequency distribution for soil temperature include the period from September 2014 through September 2016 which includes the deep peat heating period.  Data in the frequency distributions for air temperature and relative humidity include data from August 2015 through September 2016.**

**New Figure 10 is placed on Page 26 and lines 602 to 608 of the revised manuscript. Figures throughout the manuscript were renumbered as needed.**

**These data show that the overall distribution of temperatures is largely retained under the warming scenarios, but warm plot relative humidity is constrained for the warmer treatments. Because absolute humidity is not modified in the treatments, this is an expected result because of the increase in saturation mixing ratio with temperature.  This means that changes in absolute humidity (driven by weather conditions) contribute to smaller changes in relative humidity at higher temperature levels.**

It is important to present at least some the data in a manner that does not just show that the treatments are different on average, for narrow bands of time. Rather I would like to see some exploration of the daily and annual patterns observed versus what we would expect to see.

**Annual patterns of the observed absolute mean daily data for air temperature and soil temperature are already plotted in the upper graphs of Figures 3 and 4, respectively. These figures include all dates for each individual enclosure. We had not previously presented figures from the half-hour data set (Hanson et al. 2015) because they overwhelm the capacity of our graphics program. New figures were prepared as requested including: Figures 8 on line 555, Figure 9 on line 571, and Figure S6 on line 1478. Section 3.3 (lines 524 to 615) was fully revised to contain the following material.**

**The following graph (New Figure 8 added to Section 3.3.2) shows a week of the half-hourly observations by SPRUCE treatment for a summer period in late August 2015 when annual temperatures were at their maximum annual values. Data for relative humidity and air temperature at +2 m and soil temperatures at the control depth of -2 m are shown.**

[Figure]

The next graph (New Figure 9 added to Section 3.3.2) shows a week of the half-hourly observations by SPRUCE treatment for a winter period in January 2016 when annual temperatures were at their minimum annual values.

[Figure]

For both summer and winter conditions you can see that the SPRUCE system is capable of sustaining differential temperatures throughout diurnal cycles in a very consistent manner as was the case for Figures 3 and 4 of the paper. Relative humidity which is reduced with warming (see also manuscript Table 4) also follows the diurnal patterns with treatment.

Away from the active control positions (+2 m for air temperature; -2 m for soil temperature) it is important to point out that the stratification is similar, but not always maintained. The following figure for soil temperature at -10 cm (a new Figure S6 was added to the Supplemental Material; line 1478), clearly show that the treatments are largely maintained up through the soil profile (see also manuscript Figure 5), but that some differences can develop driven by the unique energy balance relationships for a given SPRUCE enclosure. Such differences are driven by variable

**tree-cover conditions that effects local energy balance responsible for the development of soil profile temperature differentials above the -2 m control depth.**

[Figure]

The previous examples of diurnal temperature patterns and differentials werel added into Section 3.3.

As this data is currently presented, there is strong difference in the daily averages of each treatment and they seem to be consistent throughout the year. But these data lump seasonal and diurnal variability and may mask patterns of efficacy that are important for the reader to understand.

**Seasonal variability is already presented in Figures 3 and 4 and Table 3, and diurnal variation is characterized in Table 2. Examples of diurnal variation across treatments are shown in the newly drafted figures above. Real-time and archived SPRUCE data are also available for consideration at http://sprucedata.ornl.gov/vdv.**

**This web site is cited in the paper on line 482.**

The authors should use the delta from ambient as a measure of the experiment look at the average and variability across various ecological scales. Hour of day (not just an individual day) would be the most important but also by time of year.

**On this point we disagree. The fully-constructed-control enclosures include shading effects and internal turbulence (as described in the paper) that need to be considered when contrasted with warmed plots (+2.25, +4.5, +6.75, and +9 C). For belowground studies, one might rationalize that the ambient plots (Plot 7 and 21) can be interpreted as another treatment level. In completed response papers we have been characterizing them as -2 °C plots (e.g., Wilson et al. 2016).** Hour-of-day data are described with representative plots in the previous answers.

Wilson RM, Hopple AH, Tfaily MM, Sebestyen S, Schadt CW, Pfeifer-Meister L, Medvedeff C, McFarlane K, Kostka JE, Kolton M, Kolka R, Kluber L, Keller J, Guilderson T, Griffiths N, Chanton JP, Bridgham S, Hanson PJ (2016) Stability of peatland carbon to rising temperatures. *Nature Communications* 7:13723, doi: 10.1038/NCOMMS13723.

The limited number of sensors makes spatial variability harder to explore in this manner but it would be important as well.

**Spatial variation was an important consideration during the development of the belowground and air warming protocols (Barbier et al. 2012) during construction and testing of the full size prototype in Oak Ridge, TN. In that system, a 3D-monitoring approach included a central tower and spaced sensors located at various heights and distances from the center of the plot. They were established and monitored to capture spatial details. During prototype development, we also monitored soil temperatures to -2 m along a radius from edge to center of the plot in that prototype. Results from the Barbier et al. (2012) paper demonstrated little spatial variation belowground, and some variable aboveground spatial homogeneity driven by external wind velocities. The greatest variation in the warm air envelope above ground occurred under calm conditions, and a full discussion of spatial considerations is included in Barbier et al. (2012).**

These details have been added to the discussion on lines 787 to 797.

I would also like to see the overall distribution of temperatures for each of the treatments. It is important that the distribution of temperatures match ambient as much as possible along with differing in mean.

New Figure 10 line 602 was drafted for this response. **The distributions revealed a very good representation of the ambient distributions for soil and air temperatures, but a somewhat constrained distribution for relative humidity as the warming treatments increase. Such variation is inherent to the experimental system. No attempt to correct this small change was attempted because there is not consistent guidance from climate models as to the exact nature of such distributions to expect for future climates.**

Some of the papers they reviewed in this ms use analyses like those suggested, I would also the see if there are seasonal patterns as well. It is easier to use the deltas for these analysis then the overall temperature.

**See our responses above.**

It is likely that variability in treatment is higher in parts of the day or times of the year and that would be important to know.

**As demonstrated in the figures above, this is typically not the case near the control points, but is inevitable as you move up the soil profile away from the -2 m controlled zone.**

I would also like to see multivariable traces and deltas for 10 days or at an hourly scale. This could be in the supplement and help the reader see the efficacy of the experiment in an ecological context.

**Half-hour data are provided in the figures above and were included in the paper (Figures 8, 9 and S6).**

It is probably beyond the scope of this paper but I would like to see an analysis linking directly the specific temperature/ light and rh conditions of sampling area with measurements just in thoses areas.

**To the extent that we could afford sensors throughout the enclosures, they have been added to allow individual tasks to associate their task-specific observations to the most appropriate environmental sensors. Temperature and relative humidity data are available at 0.5, 1, 2 and 4 m to allow canopy responses for surface vegetation, shrubs and *Picea* and *Larix* foliage to be most appropriately represented by their actual growth conditions. Due to good mixing, there isn't a lot of difference (see the lower graph - Figure 5). Soil temperature data are available at 0, -5, -10, -20, -30, -40, -50, -100 and -200 cm to allow peat and microbe response analyses to be appropriately characterized by depth.  In addition, soil temperature is assessed along these depths at three different zones within the plot to allow us to associate measurements to the closest appropriate zone. All of these data are available to project members during active operation, and through the public archive (Hanson et al. 2016) for future analyses. The paper was not, however, expanded more to include these details directly.**

I am not sure what spatial data is available but it would reassure readers to know that the sampled area variability is minimized.

**Answer repeated from above:  Spatial variation was an important consideration during the development of the belowground and air warming protocols (Barbier et al. 2012) during construction and testing of the full size prototype in Oak Ridge, TN.  In that system, a 3D-monitoring approach included a central tower and spaced sensors located at various heights and distances from the center of the plot. They were established and monitored to capture spatial details. During prototype development, we also monitored soil temperatures to -2 m along a radius from edge to center of the**

**plot in that prototype. Results from the Barbier et al. (2012) paper demonstrated little spatial variation belowground, and some variable aboveground spatial homogeneity driven by external wind velocities. The greatest variation in the warm air envelope above ground occurred under calm conditions, and a full discussion of spatial considerations is included in Barbier et al. (2012).**

**These details have been added to the discussion on lines 787 to 797.**

As it is a whole ecosystem model with some range in values, it would be nice to know whether the sampling area occupies that entire range or is experimenting a narrower range of treatments. For example, it would be great if RH decline with temperature in areas sampled was less than chamber level.

**As described above relative humidity is assessed at 0.5, 1, 2 and 4 m above the ground to provide such data. Due to good mixing within the enclosure (Barbier et al. 2012) a horizontal array of such sensors was deemed unnecessary. Of course, more data are always useful, and users of SPRUCE may add other localized sensors.**

I am especially concerned about pattern of nighttime temperature with distance from chamber wall and RH variation with distance from blower manifold.

**See the Barbier et al. (2012) paper and previous answers. Through additional spot checks, but not automated and continuous measurements, we have demonstrated that the warm air leaving the 8 source diffusers on each of the enclosure walls becomes well mixed very quickly. Nonetheless measures of shrub or tree canopies directly impacted by the source warm air are avoided and minimized.**

**No changes to the manuscript.**

There is very little discussion of soil temperature behavior during freezing and thawing cycles or by depth. This need to be include somewhere.

**Section 3.5 of the paper and the original Figure 11 (Now Figure 14), together with the modeling of ice development in Section 4.2.4 and Supplemental Figures S8, S9 and S10 cover this issue in some detail.**

I expect that soil and air temperatures invert at some point during the year and it might be better analyze these data separately.

**This is true, and it is captured in the archived data base. Future model-data intercomparison exercises underway may choose to look specifically at this phenomenon, but they are not added here to manage the length of the paper. It is important to point out, however, that such phenomenon occur in zones of the enclosures that develop their patterns due to natural energy balance phenomenon that are not impacted by the active control of deep soil temperatures. As applied, our system only produces a modified deep soil temperature to simulate future deep temperatures to be achieved with climate warming. Soil temperature patterns**

**exhibited on diurnal and annual time steps are the result of natural energy balance changes through time.**

**No changes to the manuscript.**

Again a delta based analysis of soil temperature differences would be better to show treatment effects compared to ambients rather that overall temperatures.
**Figures 3 and 4 include the differentials together with the absolute temperature values.**

**No changes to the manuscript.**

It would be nice to know that the delta variability at each depth was comparable with ambient.
**Examples of such data were provided in Figure 5, and we have added the -10 cm soil temperature Figure S6 presented earlier for this purpose. With millions of data points, we have tried to choose wisely to present the data of use to the most readers.**

Daily pattern in RH would also be nice to know as well.
**Example data have been graphed and are provided above (new Figures 8, 9 and S6).**

Line 211- Is this really 12-18m deep below wetland. Please check.
**Yes.  The helical piles needed to be driven very deep to meet the engineering requirements for stability over a decade of operation.**

**No changes to the manuscript.**

The figure sharpness seemed lacking throughout, I assume that will be corrected. I like figures with sd bars rather than separate symbols
**Figures have been resaved at high resolution and will be uploaded when authorized.**

Table 2 explanation was confusing to me.
**The wording of the Table 2 caption has been modified to clarify the content of the Table.**

Soil moisture data to back up discussion of RH and ET?
**A discussion of peat moisture content and water table data was provided in the Response to Reviewer #1, and Figure S2 demonstrating peat moisture variation in the peatland hollows was added to the Supplemental Materials section.**

Figure 7 is good. I would like to see more like this. I would like to see the same analyses for differing sensor variables. One could be in paper and other in supplement.
**We have constructed a revised Figure 7 for use in Section 3.3 line 531. It is reproduced on the next page with the related AT data as a replacement for the current Figure 7. For soil temperatures, there is essentially no variation at the sub-half hour time step**

and we have not provided those data in a figure, but the data are recorded and available within in Hanson et al. (2016). In the case of RH data below, the sub-half hour variation does not increase with warming treatment.

[Figure]

**Figure 7: Sub-half hour variation of air temperature (upper graph) and relative humidity (lower graph) data expressed as the standard deviation (SD or sd) of 1-min observations within a half hour measurement period. Plotted data are the mean SD±sd and maximum SD for half-hour temperature and relative humidity data over the whole-ecosystem-warming period of observations reported in this paper for two replicate sensors in each treatment enclosure or plot. The -2 and 0 °C treatments in this graph represent unchambered ambient and no-energy-added control enclosures respectively.**

Right now the comparison discussion between this and other warming experiment seems underdeveloped. I suggest picking a couple of key comparisons to develop discussion.

**The overall goal of this paper is to document the capacities of the SPRUCE enclosure system.  As we stated in Section 4.1 other studies have provided an in depth discussion of the advantages and disadvantageous of other approaches (Aronson and McNulty 2009, Amthor et al. 2010, Kimball 2011, LeCain et al. 2015), and we didn't choose to provide a comprehensive point-by-point comparison that would lengthen an already long paper. Rather, we wanted to provide data in Table 6 to describe the breadth of available approaches to make the point that other options are available for other ecosystems and questions.**

Temporal pattern, dewpoint, soil moisture and RH be interesting to include more of. What should RH and dewpoint looked in a good manipulation?

**Example data have been graphed and are provided above (new Figures 8, 9 and S6), and we have discussed the implications for the design on dewpoint formation (Section 4.2.5).  All data are archived for in-depth future analyses (Hanson et al. 2016).**

The summary table also need to be checked. The data from at least one of these papers is incorrect.

**Unfortunately, we have not found the error that the reviewer located.  We would be happy to make a change or changes if specific adjustments can be suggested.**

[revised manuscript text omitted]

**Comment [Office9]:** This supplemental figure was added in response to detail reqeuested by Reviewer #2.

[Figure]

**Figure S7**: Absolute humidity by treatment enclosure from mid-year 2015 through early 2016.
For clarity of the image, standard error bars all in grey are included only for the control (T+0)
and the warmest (T+9) plots.

[Figure]

**Figure S8:** Images of snow accumulation at unchambered ambient locations and within all
treatment enclosures by target warming temperature differentials at 10:00 on 6 April 2016. Little
obvious snow accumulation is apparent above the +4.5 °C treatment, even though precipitation
in the form of snow does enter all enclosures.

**Additional graphics from the SPRUCE Enclosure Energy Simulations (D. Ricciuto)**

[Figure]

**Figure S9**: Simulations of snow depth for ambient conditions (black) and within an enclosure
(grey) using driver meteorology data from 2013.

[Figure]

**Figure S10**: Profiles of simulated top 1m soil temperature in ambient (a) and enclosure (b)
simulations. Contour colors represent peat temperatures in degrees kelvin, and the black contour
indicates those layers that are below freezing during the year. Ice depths are similar between the
simulations.

**Elevated $CO_2$ Protocol Details**
During the period from January through March 2016 when biological activities were minimal,
various test were conducted on Plot 19 (a constructed control), Plot 11 (+2.25 °C), Plot 4 (+4.5
°C), Plot 8 (+6.75 °C) and Plot 10 (+9 °C) to establish the $CO_2$ addition control protocols. Over a
multi-day period with variable winds, a fixed amount of $CO_2$ ranging from 150 to 300 l min$^{-1}$ of
pure $CO_2$, depending on target temperature levels, was added to the enclosure for a multiple day
period to generate a profile of achieved $CO_2$ differentials (mean at 0.5, 1 and 2 m heights) as a
function of the wind velocities measured at +10 m. A fitted relationship between wind velocity at
+10 m and enclosure fractional air turnover volumes (assuming and enclosure volume of 911 m$^3$)
was derived from these data. Instantaneous measured wind velocities were then applied to a
turnover fraction equation to estimate the amount of $CO_2$ to be added to achieve a +500 μmol
mol$^{-1}$ value over ambient-CO2 measured within the constructed control plot (i.e., Plot 6). An
example is as follows:
TF = (0.00001330297 *WS^6) + (-0.0003804215 *WS^5) + (0.003932579 * WS^4) +
(-0.01517648 * WS^3) + (-0.004974471 * WS^2) + (0.2532064 * WS)
where TF is enclosure turnover fraction (unit less), and WS is wind velocity (m s$^{-1}$). The form of
the TF equation might also be a simple exponential function depending on the calibration data
set for a given plot.
Using the TF value, an initial coarse control value for $CO_2$ addition was calculated as:
Course $CO_2$ Addition = CCO2 = EV * TF * DetaCO2 * 1000
where CCCCO2 is the $CO_2$ addition rate in l min$^{-1}$, EV is the enclosure volume in m3 (~910 m3),
DeltaCO$_2$ is the desired target increase in $CO_2$ above ambient conditions (500 μmol mol$^{-1}$ or
0.0005 m$^3$ m$^{-3}$), and 1000 allows for the conversion from m$^3$ to liters. To further account for the
variation in enclosure turnover times with external winds the DeltaCO2 values were
supplemented with added amounts as shown in the following table.
Table S4. DeltaCO$_2$ adjustment values for low, medium and high winds by treatment plot.

| $CO_2$ Treatment Plot # | Low Wind Adjustment (ppm) | Medium Wind Adjustment (ppm) | High Wind Adjustment (ppm) |
|---|---|---|---|
| 4 | 50 | 50 | 50 |
| 10 | 125 | 75 | 40 |
| 11 | 75 | 75 | 75 |
| 16 | 50 | 25 | 0 |
| 19 | 75 | 50 | 0 |

Yet additional fine control to achieve target differential $CO_2$ concentrations within the enclosure
was based on a feedback adjustment defined by the error in achieving +500 μmol mol$^{-1}$.
CO2ERR = 500 – (CO2Enclosure – CO2Ambient)
Final $CO_2$ Addition = FCO2 = (910.6 * CO2ERR)/1000000*1000*1.15
where CO2ERR is the observed difference of enclosure $CO_2$ when compared with $CO_2$ in the
constructed control (Plot 6), 1000000 and 1000 convert m$^3$ to L, and 1.15 is an arbitrary valued
needed to achieve good results (probably accounting for unmeasured vertical winds). This combined control algorithm reevaluated every 10 seconds during active $CO_2$ additions, allowed
us to achieve target $CO_2$ levels within the enclosure within a ± 50 µmol mol$^{-1}$ band around our
target of + 500 µmol mol$^{-1}$ $CO_2$. We will continue to adjust the algorithm for $CO_2$ additions as
we operate to allow each enclosure to achieve +500 ± 25 µmol mol$^{-1}$ for all wind conditions and
temperature treatments.
Elevated $CO_2$ additions are only made during daytime hours as a cost reducing measure, because
past studies have shown that there is no direct effect of elevated $CO_2$ on respiratory processes
(Amthor 2000; Amthor et al. 2001; Tjoelker et al. 2001). The elevated $CO_2$ treatments are
initiated or stopped each day based on calculated solar angles for each day of the year using the
Solpos algorithm developed by the National Renewable Energy Laboratory (NREL).
Table S5. Mean daily differential $CO_2$ achieved from 19 August to 1 September 2016. NA = not
applicable.

| Warming Level and Plot | Differential [$CO_2$] in ppm ± sd |
|---|---|
| Reference Plot - +0.00 °C Plot 06 | NA |
| +2.25 °C Plot 20 | -9 ± 8 |
| +4.50 °C Plot 13 | -0.1 ± 8 |
| +6.75 °C Plot 13 | -13 ± 9 |
| +9.00 °C Plot 04 | 1 ± 11 |
| eCO$_2$ +0.00 °C Plot 19 | 483 ± 22 |
| eCO$_2$ +2.25 °C Plot 11 | 471 ± 21 |
| eCO$_2$ +4.50 °C Plot 04 | 490 ± 13 |
| eCO$_2$ +2.25 °C Plot 16 | 511± 15 |
| eCO$_2$ +9.00 °C Plot 10 | 480 ±73 |

Supplemental Literature
Amthor, J.S.: Direct effect of elevated $CO_2$ on nocturnal in situ leaf respiration in nine temperate
deciduous tree species is small. *Tree Physiol.* 20, 139-144, 2000.
Amthor, J.S., Koch, G.W., Willms, J.R., Layzell, D.B.: Leaf $O_2$ uptake in the dark is independent
of coincident $CO_2$ partial pressure. *J Exper Bot,* 52, 2235–2238, 2001.
Tjoelker, M.G., Oleksyn, J., Lee, T.D., Reich, P.B.: Direct inhibition of leaf dark respiration by
elevated CO2 is minor in 12 grassland species. *New Phytol,* 150, 419–424. doi:10.1046/j.1469-
8137.2001.00117.x, 2001.